



# Using wavelet transform to analyse on-road mobile measurements of air pollutants: a case study to evaluate vehicle emission control policies during the 2014 APEC summit

Yingruo Li[1, 2], Ziqiang Tan[1], Chunxiang Ye[1], Junxia Wang[1], Yanwen Wang[1], Yi Zhu[1], Pengfei Liang[1],
Xi Chen[1], Yanhua Fang[1], Yiqun Han[1], Qi Wang[1], Di He[3], Yao Wang[3], and Tong Zhu[1*]

[1]BIC-ESAT and SKL-ESPC, College of Environmental Sciences and Engineering, Peking University, Beijing 100871, China
[2]Institute of Urban Meteorology, China Meteorological Administration, Beijing, 100089, China
[3]Environmental Meteorology Forecast Center of Beijing-Tianjin-Hebei, China Meteorological Administration, Beijing, 100089, China

*Correspondence to: tzhu@pku.edu.cn

**Abstract**

Vehicle emissions are a major source of air pollution in urban areas, and thus greatly impact air quality in the megacity Beijing. Various vehicle emission control policies have been implemented at great cost, but there is a lack of appropriate methods to evaluate the effectiveness of such policies. Here we developed a wavelet transform method (WTM) to evaluate the

effectiveness of vehicle emission control policies during the 2014 Asia-Pacific Economic Cooperation (APEC) summit, taking advantage of high time resolution mobile measurements of NO, $NO_x$, BC, CO, $SO_2$, and $O_3$ made around the 4th Ring Road of Beijing. The WTM decomposed on-road mobile measurements into low- and high- frequency components, where the former represents immediate vehicle emissions, and the latter represents the atmospheric background in addition to accumulated on-road emissions. The high-frequency component of the WTM ($C_{H\_freq.}$), which represents the concentrations of pollutants from

vehicle emissions ($C_{veh.}$), was used to evaluate the changes in vehicle emission intensity in the full-APEC period (3–12 November 2014) relative to the pre-APEC (28 October to 2 November 2014) and post-APEC (13–22 November 2014) periods, during which different vehicle emission control policies were implemented. Our results suggest that the $C_{veh.}$ of NO, $NO_x$, BC, and CO in the full-APEC period were 19.4%, 17.7%, 0%, and 50% lower, respectively, than those in the pre-APEC period during daytime, and were 50%, 47.3%, 62.5%, and 50% lower than those in the post-APEC period during daytime. The $C_{veh.}$

of NO, $NO_x$, BC, and CO in the full-APEC period were 65.3%, 65.4%, 14.3%, and 50% lower than those in the post-APEC period during nighttime. These results indicate that the vehicle emission control policies implemented during the full-APEC period were effective. The WTM is a feasible and stable technique for signal decomposition of atmospheric measurements, and it is a useful method for the evaluation of pollution control policies.

## 1 Introduction

Due to socioeconomic development and fast urbanisation, vehicle usage in megacities has increased rapidly, resulting in increasing on-road emissions of air pollutants. Beijing is one megacity that suffers from serious air pollution (Parrish and Zhu, 2009; Kelly and Zhu, 2016), where vehicle emissions are important contributors to the concentrations of nitrogen oxides ($NO_x$ = NO + $NO_2$), black carbon (BC), carbon monoxide (CO), volatile organic compounds (VOCs), ammonia ($NH_3$), and fine particulate matter ($PM_{2.5}$) (Cao et al., 2016; Gao et al., 2018; Sun et al., 2017; Zhu et al., 2016).

To improve air quality, the Beijing municipal government has implemented a range of vehicle emission control policies since the 2000s, with more comprehensive policies implemented since the 2008 Olympic Games (He et al., 2010; Wang et al., 2009; Wang et al., 2010). These policies have improved the air quality of Beijing to some extent, but have also brought inconvenience to residents and high economic and social costs. Evaluation of the effectiveness of vehicle emission control policies is


important in air pollution control (Kelly and Zhu, 2016), but appropriate methods are still quite limited (Bukowiecki et al., 2002; Cheng et al., 2012; Riley et al., 2014; Liang et al., 2017; Zhou et al., 2009).

On-road mobile measurements can be used to obtain the spatial and temporal distributions of air pollutants with high time resolution. They have been widely used in the evaluation of air pollution control policies (Wang et al., 2009), and regional
transport into megacities and over a large-scale area (Wang et al., 2011; Zhu et al., 2016). These on-road measurements sampled air pollutants originating from both immediate on-road emissions and their backgrounds (atmospheric background in addition to accumulated on-road emissions). The immediate on-road vehicle emissions result in narrow and sharp spikes (i.e., high-frequency variations) in time series of pollutant concentrations, which we hereafter refer to as the concentrations from immediate vehicle emissions ($C_{\text{veh.}}$). Atmospheric background and accumulated vehicle emissions also contribute to variations
in time series of pollutant concentrations, with characteristic broad and low peaks (i.e., low-frequency variations), which we hereafter refer to as to as on-road background concentrations ($C_{\text{bg.}}$). Therefore, finding an effective method to separate the high- and low-frequency parts may help us to evaluate vehicle emission control policies.

Some efforts have been made to separate the $C_{\text{veh.}}$ and $C_{\text{bg.}}$ components of on-road mobile measurements, including but not limited to: comparing the concentrations of pollutants measured on a major road with suburban/urban background observations
(Riley et al., 2014), using the low percentiles of on-road pollutant concentrations to estimate $C_{\text{bg.}}$ (Bukowiecki et al., 2002), and using model simulations and source apportionment methods to estimate $C_{\text{veh.}}$ (Thornhill et al., 2010; Cheng et al., 2013). However, these methods usually have large uncertainties due to the spatial and temporal mismatch of on-road measurements and ground-based site measurements, reliance on subjective judgement for the selection of appropriate time windows and percentiles, and inherent uncertainties in model construction and inputs.
The wavelet transform (WT) is a time–frequency analysis method developed from the Fourier transform (FT) (Daubechies et al., 2015; Domingues et al., 2005). The FT allows for the transformation of signals from the time to the frequency domain. However, in the FT process, time information that may be quite important is discarded. Therefore, the short-term Fourier transform (STFT) was developed for transforming signals from the time to the time–frequency domain with uniform time–frequency windows to preserve such time information (Avargel et al., 2007). However, the WT allows for variable time and
frequency windows, which are adjusted according to dynamic changes in the signals during signal processing. The WT is a powerful technique for time–frequency analysis of non-stationary signals. It has a wide range of applications in image processing, signal processing (e.g., monitoring signal discontinuity), identifying signal trends, and achieving noise reduction (Akansu et al., 2010; Daubechies et al., 2015; Domingues et al., 2005). The WT has also been successfully applied to many problems in the field of atmospheric science (Dunea et al., 2015; Li et al., 2016; Tian et al., 2014). The pollutant concentrations
obtained by on-road mobile measurements can essentially be regarded as non-stationary signals that consist of different frequencies, as discussed above. Therefore, WTs can be used to extract the narrow and sharp spikes in time series of air pollutants by decomposing the signals of on-road mobile measurements to different frequencies, and in theory, the WT can be used to evaluate the effectiveness of vehicle emission control policies.

The 21st Asia-Pacific Economic Cooperation Forum (APEC) summit was held in Beijing from 5 to 11 November 2014. To
achieve good air quality during the APEC summit, the government implemented a series of air pollution control policies, with a particular focus on vehicle emissions (Chen et al., 2015; Liang et al., 2017; Liu et al., 2016; Wen et al., 2016). The APEC summit thus provides an ideal opportunity to evaluate the effectiveness of vehicle emission control policies. In this study, we carried out daytime and nighttime on-road mobile measurements around the 4th Ring Road of Beijing during the APEC summit period from 28 October to 22 November 2014, including the periods "pre-APEC" (28 October to 2 November 2014), "full-
APEC" (3–12 November 2014), and "post-APEC" (13–22 November 2014). Air pollutants, including nitric oxide (NO), NO$_x$, BC, CO, sulfur dioxide (SO$_2$), and ozone (O$_3$) were measured, as well as corresponding auxiliary parameters, such as the GPS information latitude and longitude, temperature ($T$), relative humidity (RH), and pressure ($P$).



We developed a wavelet transform method (WTM) based on the WT to decompose the measured pollutant concentrations into low- and high-frequency components. As mentioned above, the low-frequency component represents $C_{bg.}$. The high-frequency component represents $C_{veh.}$ and was used to evaluate the vehicle emission control policies implemented during the 2014 APEC summit. We give detailed descriptions of the experimental design and a brief introduction to the WTM in Sect. 2. In Sect. 3,

we first give an overview of the on-road mobile measurements in Sect. 3.1, and then we give the results of the WTM and compare it with other methods in Sect. 3.2. Finally, in Sect. 3.3, we used the WTM to evaluate the effectiveness of vehicle emission control policies implemented during the 2014 APEC summit in Beijing as a case study.

## 2 Experimental design and methods

### 2.1 Experimental design

The mobile measurement platform for the on-road air pollutant measurements was integrated into a diesel van (IVECO Turin V; length × width × height = 6.0 m × 2.4 m × 2.8 m), as shown in Fig. 1. The mobile measurement platform mainly consists of sampling, damping, cooling, power, and auxiliary measurement systems. Detailed information regarding the platform can be found in previous studies (Wang et al., 2009, 2011; Zhu et al., 2016). The sampling system included gaseous and particle sampling systems. The gaseous sampling system used a pump and two mass flow controllers to ensure a stable airflow

unaffected by vehicle driving state. The sampling inlet was located at the front part of the platform to reduce sampling of the van exhaust. The damping system was designed to prevent damage to the instruments during driving and ensure stable measurements and data transmission. The cooling system was used to keep the temperature inside the van constant. The power system consisted of six sets of 48 V/110 Ah lithium battery packs and two sets of uninterruptible power supplies (UPSs). Each set of UPSs controlled three sets of battery packs to allow for 5 h of uninterrupted measurements. The auxiliary measurement

system was used to collect GPS, traffic condition, and meteorological data. Concentrations of $SO_2$, NO, $NO_x$, $O_3$, CO, BC, and the auxiliary data, including latitude and longitude, $T$, RH, and $P$, were collected in this study. Details of the instruments onboard are referred to pervious studies (Wang et al., 2009, 2011; Zhu et al., 2016). The gas analysers for NO–$NO_x$, CO, $SO_2$, and $O_3$ were calibrated once a week and more frequently when necessary. Occasionally, instrument failure resulted in missing data for NO, $NO_x$, BC, and RH.

The on-road measurement route is shown in Fig. 1. For each experiment, the van started from the Peking University (PKU), and drove around the 4th Ring Road of Beijing (circumference ~ 65 km) and then returned to PKU. The driving routes were always counter-clockwise except for 28 October 2014. The 4th Ring Road of Beijing was chosen to carry out the experiment for the following reasons. First, PKU is close to the 4th Ring Road, which is convenient for vehicle dispatch. Second, the 4th Ring Road is the boundary of the central and suburban districts of Beijing and also an important reference line for vehicle

restriction policies. For example, vehicle licence plate restrictions were implemented within but not including the 5th Ring Road of Beijing from 07:00 to 20:00 local time (LT) every weekday. Finally, the 4th Ring Road is an important and representative main road of Beijing.

Figure 1. Here

On-road measurements were carried out around the 4th Ring Road of Beijing from 28 October to 22 November 2014. The

APEC summit was held in Beijing from 5 to 11 November 2014. The Chinese government implemented a series of air pollution control policies to ensure good air quality in Beijing during the APEC summit. The Ministry of Environmental Protection of China formulated "the assurance program of air quality during the APEC summit". The area covered by air pollution control policies included Beijing, Tianjin, Hebei, Shanxi, Inner Mongolia, and Shandong. The air pollution control policies were implemented in a stepwise manner before the APEC summit commenced. In this study, the observation period (28 October to

22 November 2014) was divided into three time periods according to when air pollution control policies were implemented, with a particular focus on the odd–even vehicle licence plate rule (i.e., only vehicles with odd last digits on their licence plates



were allowed to drive on one day, and only those with even last digits on the next day, and so on), imposed during the APEC summit:

1. *Pre-APEC (28 October to 2 November 2014)*. During this period, control policies on vehicles and coal combustion were implemented gradually. An early warning system for air quality was established, with a series of contingency plans developed that would be put in place should heavy pollution episodes occur during the 2014 APEC summit.

2. *Full-APEC (3–12 November 2014)*. During this period, air pollution control policies on coal combustion, industrial, dust, and vehicle emissions were fully implemented. Some heavily-polluting industrial factories were shut down. Strict temporary vehicle emission control policies and the odd–even vehicle licence plate rule were also imposed during this period.

3. *Post-APEC (13–22 November 2014)*. During this period, the odd–even vehicle licence plate rule was lifted and the normal licence plate restriction policy was resumed. Industrial factories gradually resumed production.

Further details of these air pollution control policies are given in Table 1.

Table 1. Here

Over the 26 days of observations, on-road mobile measurements were conducted during 44 circuits of the 4th Ring Road (Table S1). The morning observation period was from 09:50 to 12:00 LT, and the nighttime from 01:00 to 02:30 LT. The observation vehicle started around 10:00 LT in the morning, mainly for better development of the atmospheric boundary layer and the reduced traffic congestion during this time (i.e., after the morning rush hour) (Tao et al., 2007; Han et al., 2014). We also conducted on-road mobile measurements during the nighttime to explore the differences in vehicle emission control policies between daytime and nighttime (Table 1). According to Bukowiecki et al. (2002), the probability of measured results being affected by self-pollution is only 5% when the driving speed of a mobile laboratory exceeds 5 km/h. The driving speed in this study was kept at ~60 km/h to maintain a constant sampling flow rate and reduce self-pollution. The effects of self-pollution can thus be neglected for most of the observations. The total driving times for the measurements were around 1–1.5 h, except for 14, 18, 20, and 21 November 2014, during which there was substantial traffic congestion that led to longer measurement times.

## 2.2 The wavelet transform method (WTM)

The WT is a powerful technique for time–frequency analysis and is widely used in seismic data analysis, transient signal analysis, and other signal processing applications (Tian et al., 2014; Domingues et al., 2005; Kang et al., 2007). Unlike the STFT, which decomposes a signal into a set of equal-bandwidth basis functions in the spectrum, the WT provides a decomposition based on constant–Q (equal bandwidth on a logarithmic scale) basis functions with improved multi-resolution characteristics in the time–frequency plane (Akansu et al., 2010). The WT maps the function $f(t)$ in $L^2(R)$ to another signal $W_f$ $(a, b)$ in $L^2(R^2)$, where $a$ and $b$ are continuous and referred to as the scaling and shift parameters, respectively. The continuous WT of a time series $f(t)$ is defined as:

$$W_f(a, b) = \{f(t), \psi_{ab}(t)\} = \int_{-\infty}^{\infty} f(t)\, \bar{\psi}_{ab}(t)\mathrm{d}t \quad a > 0 , \tag{1}$$

$$\psi_{ab}(t) = \frac{1}{\sqrt{a}}\psi(\frac{t-b}{a}) , \tag{2}$$

where $f(t)$ represents the time series of a signal, $\psi(t)$ is the mother wavelet function, and $\bar{\psi}_{ab}(t)$ is the conjugate complex of $\psi_{ab}(t)$. Selecting the appropriate mother wavelet function ($\psi(t)$) is key to the WTM. The continuous scaling and shift parameters ($a, b$) make the continuous WT quite redundant and result in high computational cost. Thus, the discrete WT, which uses discrete values of scale ($j$) and localization ($k$), was used instead. If we let the sampling grid $a = a_0^{\ j}$ and $b = kb_0 a_0^{\ j}$, the discrete WT is defined as:

$$\{\psi_{jk}(t)\} = \psi_{jk}(t) = a_0^{\frac{-j}{2}}\psi(a_0^{-j}t - kb_0) \tag{3}$$





The commonly used mother wavelet function include Haar (denoted by "haar"), Daubechies ("dbN", where N is an integer), Coiflet ("coifN"), Symlets ("symN"), Morlet ("morl"), Mexican Hat ("mexh"), and Meyer ("meyr") (Fig. S1). In this study, the Daubechies wavelet functions ("dbN") were chosen as mother wavelet functions to conduct the discrete WT, mainly based on the similarity principle (Daubechies et al., 1992; Sang and Wang, 2008; Thornhill et al., 2010) (Fig. S1). The number of

5 decomposition levels was set in the range 5–9 based on considerations of computational costs, information integrity, and boundary effects. A sensitivity test was conducted to investigate how the parameters, such as the mother wavelet function and decomposition level, affected the results of the WTM. MATLAB (R2016a) was used to conduct the WT decomposition.

After WT decomposition, the original signals (measured on-road pollutant concentrations in this study) were divided into low- and high-frequency components that reflect the approximated and detailed information of the original signals, respectively

(Fig. S2). The results of the WT decomposition included both positive and negative values for the high-frequency component, which were due to the mother wavelet vibrating near zero. To reconcile this problem, the high-frequency component was moved up using a REF-line (i.e., reference, based on a running 5 min minimum of the raw data) and the low-frequency component was moved down in a homologous manner (Fig. S3). Through WT decomposition and REF-line adjustment, we obtained the results of the WTM as follows:

$$C_{\text{Total}} = C_{\text{L\_freq.}} + C_{\text{H\_freq.}} = C_{\text{bg.}} + C_{\text{veh.}} \,, \tag{4}$$

where $C_{\text{Total}}$ is the measured on-road pollutant concentrations and corresponds to their original signals, and $C_{\text{L\_freq.}}$ and $C_{\text{H\_freq.}}$ are the low- and high-frequency components of the WTM, respectively. For the pollutants typically related to vehicle emissions, such as NO, $NO_x$, BC, and CO, $C_{\text{L\_freq.}}$ is equivalent to their on-road background concentrations ($C_{\text{bg.}}$) and $C_{\text{H\_freq.}}$ is equivalent to the contributions from immediate vehicle emissions ($C_{\text{veh.}}$). The vehicle emission control policies implemented during the

20 full-APEC period could have affected both traffic flow and the vehicle fleet profile (which could change emission intensities), and thus could influence the contributions of vehicle emissions to pollutant concentrations. Therefore, the high-frequency component of the WTM can be used to evaluate the vehicle emission control policies.

To verify the suitability and efficacy of the WTM in the signal decomposition of on-road measured pollutant concentrations, the results of the WTM were compared with the moving low percentile method (MLPM), observations at the PKU site, and a

25 previous study.

## 3 Results and discussion

### 3.1 Results of on-road mobile measurements

We give an overview of the results of the on-road mobile measurements in this section. Both meteorological conditions and the temporal and spatial distributions of air pollutants were investigated.

Meteorological conditions, especially wind direction (WD) and speed (WS), play key roles in the dispersal and transport of air pollutants. Comprehensive understanding of the transport of air pollutants between megacities and their surrounding areas and separation of the contributions of meteorological conditions and air pollution control policies is important for the objective evaluation of such control policies (Li et al., 2016, 2019; Liang et al., 2017). In this study, we assumed that the high-frequency component is confined to much shorter time scales than those of the meteorological factors, and that changes in meteorological

conditions were reflected only in the low-frequency component, such that changes in meteorological conditions did not influence our evaluation of control polices.

As shown in Fig. 2, the meteorological conditions (measured at the Nanjiao site, see Fig. 1) varied over the course of the measurement period. The average wind speed during the pre-APEC period was 1.8 m s$^{-1}$, and there were strong contributions from the northwest (~25%) and northeast (~30%) wind sectors. The average wind speed during the full-APEC period was 2.1

40 m s$^{-1}$, which is higher than that during the pre- and post-APEC periods. The prevailing winds during the full-APEC period



were from the northwest (~20%) and southwest (~25%). The average wind speed for the post-APEC period was 1.5 m s$^{-1}$, which is the lowest of the three periods. The prevailing winds during the post-APEC period were from the northeast (~35%) and southwest (~25%). For Beijing, the northwest sector generally corresponds to clean air masses, while the northeast and south wind sectors have many hotspots of air pollutant emissions (Wu et al., 2011; Lin et al., 2011; Li et al., 2016). Thus, the

WD and WS during the post-APEC period were less favourable for the dispersal of air pollutants than during the other two periods, whereas the dispersion and transport conditions during the pre- and full-APEC periods were virtually indistinguishable.

Figure 2. Here

As shown in Fig. 2, RH was higher during the full- and post-APEC periods relative to the pre-APEC period, especially at night. $T$ decreased significantly after 6 November 2014, with a maximum of < 15°C in the daytime and a minimum of ~5°C at night.

Similar to WD and WS, RH and $T$ during the post-APEC period were less conducive to good air quality than during the other two periods. High RH may facilitate some liquid phase and heterogeneous chemical reactions (Yu et al., 2018; Zhu et al., 2011; Zhao et al., 2018). Meanwhile, the sudden drop in temperature may have led to an increase in heating demand, which could have had an impact on household pollutant emissions (Liu et al., 2016).

Figure 3. Here

Figure 3 shows time series of NO, NO$_x$, BC, CO, SO$_2$, and O$_3$ observations from 28 October to 22 November 2014, for which the mean ± 1 standard deviation (SD) concentrations were 337.7 ± 330.4 ppb, 416.6 ± 373.2 ppb, 4.3 ± 3.7 μg m$^{-3}$, 2.1 ± 1.6 ppm, 11.8 ± 9.5 ppb, and 5.0 ± 5.0 ppb, respectively (Table 2). The measured concentrations are comparable to previous observations in China but are higher than those in Europe and America (Zhu et al., 2016; Hagemann et al., 2014; Padró-Martínez et al., 2012; Westerdahl et al., 2005). For example, on-road mobile measurements in the North China Plain (NCP)

during summer 2013 yielded mean NO$_x$, BC, CO, and SO$_2$ concentrations of 422 ppb, 5.8 μg m$^{-3}$, 1006 ppb, and 15 ppb, respectively (Zhu et al., 2016). The mean concentrations of NO$_x$ for mobile measurements made during 2010 in Karlsruhe, Germany, were 20 ppb in inner-city areas and 30 ppb in traffic-influenced locations (Hagemann et al., 2014). The measured average on-road concentrations of NO, NO$_x$, BC, and CO were about 30 ppb, 50 ppb, 1 μg m$^{-3}$, and 0.5 ppm, respectively, in Somerville, USA during 2009–2010 (Padró-Martínez et al., 2012). The average measured on-road concentrations of NO$_x$ were

140 and 230–470 ppb on arterial roads and freeways, respectively in Los Angeles, USA, during spring 2003 (Westerdahl et al., 2005).

The mean ± 1 SD concentrations of NO, NO$_x$, BC, CO, SO$_2$, and O$_3$ during the full-APEC period were 207.3 ± 135.7 ppb, 262.7 ± 151.4 ppb, 2.3 ± 2.4 μg m$^{-3}$, 1.3 ± 0.9 ppm, 6.1 ± 4.3 ppb, and 6.8 ± 6.5 ppb, respectively, which reflect substantial decreases compared to those in the pre-APEC period, and are much lower than those in the post-APEC period (Table 2). The

decreases in pollutant concentrations during the full-APEC period are partially attributed to the implementation of air pollution control policies, in addition to changes in local emissions (e.g., commencement of the heating season in the post-APEC period) and the regional transport of air pollutants.

Table 2. Here

Figures 4 and 5 show the spatial distributions of pollutants around the 4th Ring Road of Beijing during the observation period.

In general, for NO, NO$_x$, and CO, the spatial distributions were non-uniform with several short-term spikes observed during circuits of the ring road. The spatial distributions of BC, SO$_2$, and O$_3$ were relatively uniform compared to those of NO, NO$_x$, and CO.

Figure 4. Here

The measured on-road concentrations of NO (Fig. 4) and NO$_x$ (data not shown) were > 500 ppb on some polluted days. These

high values were often found in the southern part of the 4th Ring Road and were observed during both daytime and nighttime. For example, during the nights of 14, 15, 18, and 21 November 2014, the concentrations of NO and NO$_x$ in the eastern part of the Ring Road were > 500 ppb, and sometimes even > 1000 ppb. During the nights of 16 and 20 November 2014 and in the



daytime on 16 November 2014, the concentrations of NO and $NO_x$ in the western part of the Ring Road were also high, while their concentrations in the northwestern part were relatively low.

Figure 5. Here

Three days, namely 31 October, 5 November, and 15 November 2014, were selected to explore the temporal and spatial
distributions of air pollutants (Fig. 5). The measured on-road concentrations of NO, $NO_x$, BC, and CO were higher during nighttime than during daytime. The measured on-road concentrations of BC increased to > 10 μg m$^{-3}$, and even 20 μg m$^{-3}$, in some parts of the 4th Ring Road of Beijing during the post-APEC period, such as during the nights of 15–21 November 2014 and in the daytime on 16, 19, and 20 November 2014. High BC concentrations were often found in the southeastern part of the Ring Road. From 18 to 21 November 2014, the concentration of on-road measured CO reached up to 10 ppm in some parts
of the Ring Road. The concentration of CO was generally higher in the southern part than the northern part of the Ring Road. High concentrations also appeared in the northwest and northeast. The measured on-road concentration of $SO_2$ sometimes increased to above 20 ppb during the post-APEC period, such as during the nights of 15–16 and 18–21 November 2014 and in the daytime on 16 and 18–20 November 2014. In contrast to other pollutants, there was no significant difference in the concentrations of $SO_2$ between daytime and nighttime. The spatial distribution of $SO_2$ shows more obvious regional
characteristics, with a lack of short-term peaks in its concentration. The temporal distribution of $O_3$ was quite different from other pollutants, in that its concentration increased significantly during the full-APEC period. During the nighttime of 6 November 2014 and in the daytime on 2 and 12 November 2014, the on-road concentrations of $O_3$ increased to > 20 ppb. In general, daytime $O_3$ concentrations were higher than those during nighttime. High concentrations of $O_3$ were often found in the northeastern, southeastern, and southwestern parts of the Ring Road.
We obtained the spatial distribution characteristics of pollutants around the 4th Ring Road of Beijing using on-road mobile measurements. The concentrations of NO, $NO_x$, BC, CO, and $SO_2$ during the full-APEC period were lower than those during the pre-APEC period and substantially lower than the post-APEC period. The reasons for the dramatic increases in pollutant concentrations after the APEC summit may be attributed to: increases in emissions due to the stoppage of air pollution control policies (Table 1), increases in the energy demand for heating due to temperature reductions (Liu et al., 2016), and
unfavourable meteorological conditions (Fig. 2). The increase in $O_3$ concentrations during the full-APEC period was related to reductions in the NO-titration effect, i.e., due to decreases in NO concentrations (Wang et al., 2017). The concentrations of NO, $NO_x$, BC, and CO during nighttime were higher than those during daytime, which may be partially attributed to decreases in the boundary layer height and increased truck emissions during nighttime (Fan et al., 2016). $SO_2$ concentrations showed no significant difference between the nighttime and daytime, which is consistent with previous observations of a double peak in
the diurnal profile of $SO_2$ observations in Beijing (Lin et al., 2012).

Vehicle emissions make important contributions to the concentrations of NO, $NO_x$, BC, and CO (Zhou et al., 2012; Cheng et al., 2013), thus on-road measurements of these species have previously been used to investigate vehicle emissions (Wang et al., 2009; Zhu et al., 2016). Short-term peaks in on-road measured NO, $NO_x$, and CO concentrations reflect their immediate emissions from on-road vehicles (Westerdahl, et al., 2009; Han et al., 2014). However, in the present work, the short-term
peaks in BC and CO were not as obvious as those of NO and $NO_x$. The reasons for these differences are as follows. First, the lifetimes of NO and $NO_x$ in the atmosphere are only a few hours (Li et al., 2016), which are much shorter than those of BC and CO with lifetimes of 1 week (Samset et al., 2014) and nearly 1 month (Li et al., 2016), respectively. Therefore, NO and $NO_x$ concentrations were more sensitive to the impacts of immediate vehicle emissions compared with BC and CO. Second, the response times of instruments usually used for BC and CO measurements are longer than those for NO and $NO_x$, which
limits mobile measurements in their ability to capture instantaneous vehicle emission plumes during on-road observations.

On-road mobile measurements can be used to obtain spatial distributions of pollutants with high time-resolution over large areas. Here we investigated the spatial distributions of pollutants around the 4th Ring Road of Beijing to understand the distribution characteristics of different pollutants. According to their distribution features, we found that the typical pollutants





NO, $NO_x$, BC, and CO (NO and $NO_x$ in particular) were suitable for the evaluation of vehicle control policies in this study. We also observed differences in pollutant behaviour between daytime and nighttime.

### 3.2 Results of the WTM and comparison with other methods

To select appropriate parameters for the WTM, we chose different mother wavelet functions (db4–db8) and numbers of

decomposition levels (5–9 levels) to carry out the WT. The average high-frequency components (i.e., $C_{H\_freq.}$) of the WTM for NO, $NO_x$, BC, CO, $SO_2$, and $O_3$ using different mother wavelet functions and decomposition levels are listed in Table 3. We first conducted the WTM using a fixed mother wavelet function (db4) and varied the number decomposition levels (5–9 levels). This sensitivity test showed that the decomposition result was insensitive to the number of decomposition levels. We carefully investigated the time series of the WT results and found that an eight-level decomposition could describe the characteristics

of on-road vehicle emissions correctly (Westerdahl et al., 2009; Han et al., 2014). We then conducted the WTM with a fixed decomposition level (eight levels) and varied the mother wavelet function (db4–db8). Considering the head–tail effect of the WT, we chose db6 as the most appropriate mother wavelet function (Figs. S2 and S3). The uncertainty in the WTM for NO, $NO_x$, BC, and CO was only 0.3%, 0.3%, 1.1%, 3.2%, 2.6%, 1.4%, respectively, which highlights the stability of the WTM. Table 3. Here

Figure 6. Here

Figure 6 shows the time series obtained from the WTM for NO, $NO_x$, BC, CO, $SO_2$, and $O_3$ during the observation period, using db6 as the mother wavelet function and eight levels of decomposition. The measured on-road pollutant concentrations were decomposed into a high-frequency component ($C_{H\_freq.}$) and a low-frequency component ($C_{L\_freq.}$). For the typical pollutants NO, $NO_x$, BC, and CO, $C_{L\_freq.}$ is equivalent to $C_{bg.}$ and $C_{L\_freq.}$ is equivalent to $C_{veh.}$, as discussed previously. As

shown in Fig. 6, $C_{H\_freq.}$ could only account for a small proportion of the measured concentrations of $SO_2$ and $O_3$. The variations in $C_{H\_freq.}$ ($C_{veh.}$) for NO and $NO_x$ were the largest among the measured pollutants, whereas the variations in $C_{H\_freq.}$ ($C_{veh.}$) for CO and BC were not as significant.

To verify the accuracy and efficacy of the WTM, the following techniques were used. First, we used the MLPM to separate $C_{veh.}$ and $C_{bg.}$ (Fig. 7) and compared this to the results of the WTM. Second, $C_{L\_freq.}$ from the WTM was compared with the

pollutant concentrations observed at the PKU site (Fig. 8 and Fig. S1). Finally, we calculated the proportion of $C_{H\_freq.}$ in the WTM, and compared our results with a previous study.

The MLPM is traditionally used to separate $C_{veh.}$ and $C_{bg.}$ (Bukowiecki et al., 2002). In this study, we also used the MLPM and compared its results with the WTM. Figure 8 shows the time series of MLPM decomposition results for NO, $NO_x$, BC, and CO. $C_{bg.}$ was regarded as the 5th percentile of 5 min pollutant concentrations. $C_{veh.}$ was estimated by subtracting $C_{bg.}$ from

$C_{Total}$. Table 4 shows the results of sensitivity tests for the MLPM. The results of the WTM and the MLPM are comparable, but the sensitivity test results show that the WTM is more stable than the MLPM (Figs. 7 and 8, Tables 3 and 4). Figure 7. Here

Table 4. Here

Figure 7 shows the comparison of $C_{L\_freq.}$ obtained by the WTM with observed concentrations of NO, $NO_x$, CO, $SO_2$, and $O_3$

at the PKU site. The variations in on-road background concentrations obtained by the WTM were similar to the concentrations observed at the PKU site. The correlation coefficients ($R^2$) between $C_{L\_freq.}$ from the WTM and PKU site observations were 0.63, 0.64, 0.68, 0.88, and 0.78 for NO, $NO_x$, CO, $SO_2$, and $O_3$, respectively (Fig. S4). However, for NO and $NO_x$, the absolute values of $C_{L\_freq.}$ ($C_{bg.}$) from the WTM were higher than the concentrations observed at the PKU site. This is attributed to the following reasons. First, PKU is a fixed ground-based site with limited regional representation, but the on-road measurement

covered large areas. Second, $C_{bg.}$ obtained by the WTM for NO and $NO_x$ included both their atmospheric background concentrations and their accumulated concentrations from vehicle emissions. The above comparison indicates that the WTM could be a feasible way to separate $C_{veh.}$ from the on-road measured concentrations of air pollutants.



Figure 8. Here

The proportion of $C_{H\_freq.}$ decomposed by the WTM is listed Table 5. For the whole observation period, the mean contributions of $C_{H\_freq.}$ ($C_{veh.}$) relative to the original signals were 42% ± 51%, 36% ± 47%, and 19% ± 32% for NO, $NO_x$, and CO, respectively, with little change between the pre-APEC, full-APEC, and post-APEC periods. The proportion of $C_{L\_freq.}$ for BC

and $SO_2$ increased by ~10% during the post-APEC period relative to the pre- and full-APEC periods, which may be attributed to increases in the contributions from coal combustion emissions. It is well known that NO and $NO_x$ are strongly affected by immediate vehicle emissions, thus the variations in $C_{H\_freq.}$ ($C_{veh.}$) for NO and $NO_x$ were the largest. BC is also affected by vehicle emissions to some extent, but since the instrument response time for BC in our study was relatively long (60 s), we speculate that the decomposition results of the WTM could not fully capture the immediate on-road emission changes for BC.

Therefore, both emission sources and instrument response times have impacts on the decomposition results of the WTM.

Table 5. Here

Cheng et al. (2013) used environmental monitoring data combined with model simulations and source apportionment methods to investigate the impacts of vehicle emissions. The results showed that vehicle emissions contributed ~66% and ~27% of the concentrations of $NO_x$ and BC, respectively, for measurements made during the autumns of 2011 and 2012 in Beijing. The

proportions of $C_{veh.}$ for $NO_x$ (36%) and BC (14%) in our study are about half of the percentages calculated by Cheng et al. (2013). Such differences may be attributed to the fact that $C_{veh.}$ in our study is defined as the immediate vehicle emissions, whereas the contributions calculated by Cheng et al. (2013) also included accumulated on-road background concentrations.

### 3.3 Case study: evaluation of vehicle emission control policies

Air pollution control policies were gradually implemented prior to the APEC summit and were quickly abandoned after the

summit. Stricter vehicle emission control policies were implemented during the full-APEC period, such as the odd–even licence plate rule instead of the normal licence plate restriction policy, and limits on driving time and range were more stringent for non-local vehicles, trucks, and government cars. Table 6 shows the mean ± 1 SD $C_{veh.}$ of the WTM and the relative changes in NO, $NO_x$, BC, and CO during different observation periods. Our results show that the changes in $C_{veh.}$ of NO, $NO_x$, BC, and CO during the full-APEC period were −2.3%, −3.9%, +66.7%, and −25.0%, respectively, relative to $C_{veh.}$ in the pre-APEC

period, and −56.1%, −55.4%, −28.6%, and −40.0% relative to $C_{veh.}$ in the post-APEC period. The changes in $C_{veh.}$ of NO, $NO_x$, BC, and CO during the full-APEC period demonstrate that the vehicle emission control polices implemented for the APEC summit were successful. This can firstly be attributed to the reduced population of vehicles on road, and secondly to more fluid traffic flow in the full-APEC period relative to the pre- and post-APEC periods. The magnitudes of the decreases in $C_{veh.}$ relative to the pre-APEC period were lower than those relative to the post-APEC period, which may be due to missing

nighttime measurement data in the pre-APEC period and/or the fact that some pollution control policies were gradually implemented prior to the APEC summit.

Table 6. Here

Meanwhile, the emission control policies for cars and trucks were more stringent in the daytime than at night. Table 7 shows the changes in $C_{veh.}$ during different observation periods and between daytime and nighttime. The mean $C_{veh.}$ of NO, $NO_x$, BC,

and CO during daytime in the full-APEC period were 19.4%, 17.7%, 0, and 50% lower, respectively, than those in the pre-APEC period, and were 50%, 47.3%, 62.5%, and 50% lower than those in the post-APEC period. The magnitudes of the decreases in $C_{veh.}$ for daytime NO, $NO_x$, and BC relative to the pre-APEC period were lower than those relative to the post-APEC period, which confirms that the gradual implementation of control policies during the pre-APEC period had impacts on air quality, as discussed above. The mean $C_{veh.}$ of NO and $NO_x$ during nighttime in the full-APEC period were 65.3% and 65.4%

lower, respectively, than those in the post-APEC period. The nighttime $C_{veh.}$ of NO, $NO_x$ and CO in the full- and post-APEC periods were comparable. The magnitude of the decrease in $C_{veh.}$ for nighttime BC in the full-APEC period relative to the post-APEC period was less than that for the daytime, which is related to the differences in vehicle emission control policies between





day and night, i.e., the less stringent control policies for trucks (which run on heavy-duty diesel and emit much more BC than cars) for the hours 02:00–06:00 over the entire observation period. As shown in Figure 9, BC/CO ratios reflect changes in the fleet profile, i.e., due to the reduced controls on trucks overnight compared with other vehicles during the APEC summit.

Table 7. Here

Figure 9. Here

**Conclusions**

In this study, we developed and validated a WTM to evaluate the effectiveness of vehicle emission control policies implemented during APEC. The results of the WTM were carefully compared with observations at a ground-based site, the MLPM, and the results of a previous studies. These comparisons as well as sensitivity tests showed that the WTM is a feasible

and stable method for separation of the concentrations contributed by immediate vehicle emissions ($C_{veh.}$) and on-road background pollutant concentrations ($C_{bg.}$) and can thus be used to evaluate vehicle emission control policies.

The on-road mobile measurements could be used to obtain the temporal and spatial distributions of air pollutants and capture the characteristics of on-road vehicle emissions. The data from on-road measurements around the 4th Ring Road of Beijing from 28 October to 22 November 2014, including the pre-APEC (28 October to 2 November 2014), full-APEC (3–12

November 2014), and post-APEC (13–22 November 2014) periods, were used to conduct a case study.

The $C_{veh.}$ of NO, $NO_x$, BC, CO during daytime in the full-APEC period were 19.4%, 17.7%, 0%, and 50% lower, respectively, than those in the pre-APEC period, and were 50%, 47.3%, 62.5%, and 50% lower than those in the post-APEC period. The $C_{veh.}$ of NO, $NO_x$, BC, and CO during nighttime in the full-APEC period were 65.3%, 65.4%, 14.3%, and 50% lower than those in the post-APEC period. Overall, our results indicate that the vehicle emission control policies implemented during the

full-APEC period were effective. Further exploration of the differences in emission reductions between nighttime and daytime suggested that the effectiveness of the vehicle control policies was relatively uniform. However, the less stringent controls on trucks for the hours 02:00–06:00 resulted in less effective abatement of BC emissions, since heavy-duty diesel trucks are an important source of BC (Westerdahl et al., 2009).

However, it should be noted that this study also has some limitations. First, the instrument time resolution was not able to fully

capture the instantaneous emission signals during the on-road mobile measurements, especially for BC. Therefore, faster time response air pollution instruments are needed for future studies. Meanwhile, the measurements were only conducted around the 4th Ring Road of Beijing and thus the results are not fully representative of vehicle emissions in Beijing. Future experiments should include observations for different road conditions and different types of road to enable a more comprehensive study of the contribution of vehicle emissions to air pollution in Beijing. Finally, data on traffic flow and

vehicle fleet profiles were lacking, which will be needed to further understand the effectiveness of vehicle emission control policies. We assumed that changes in meteorological conditions were reflected in the low-frequency components and did not influence our evaluation of pollution control polices. The impact of meteorological conditions must be completely eliminated to allow for more accurate assessments of pollution control policies in future studies.

**Author contributions.** Yingruo Li and Ziqiang Tan contributed equally to the paper. Tong Zhu designed the experiments. Ziqiang Tan, Pengfei Liang, Yi Zhu, Xi Chen, Yiqun Han, Yanhua Fang, and Qi Wang carried out the experiments. Yingruo Li conducted the data analysis with contributions from all co-authors. Junxia Wang managed the data. Di He and Yao Wang provided the data for meteorological parameters at the Nanjiao site. Yingruo Li prepared the manuscript with help from Tong Zhu and Chunxiang Ye.

**Data availability.** The data for mobile and stationary measurements used in this paper are available on request.




**Acknowledgements.** This study was supported by the National Natural Science Foundation Committee of China (41421064, 21190051, 41121004, 41571130024). We are thankful to Yong Zhao who participated in the on-road measurements during APEC 2014. We acknowledge the help of Robert Woodward-Massey in finalising the manuscript.

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

40



# Figures

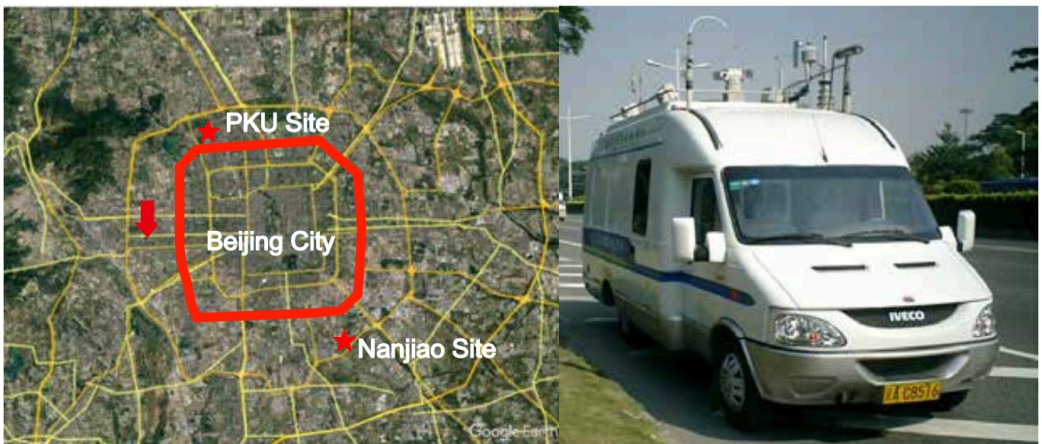

**Figure 1.** The measurement route (left) and the mobile measurement platform (right). The route started from the Peking University (PKU),
driving counter-clockwise around (indicates by the red arrow) the 4th Ring Road of Beijing (circumference ~65 km) before returning to
5   PKU. The red line represents the 4th Ring Road of Beijing, and the red stars show the ground-based observation sites (PKU and Nanjiao).

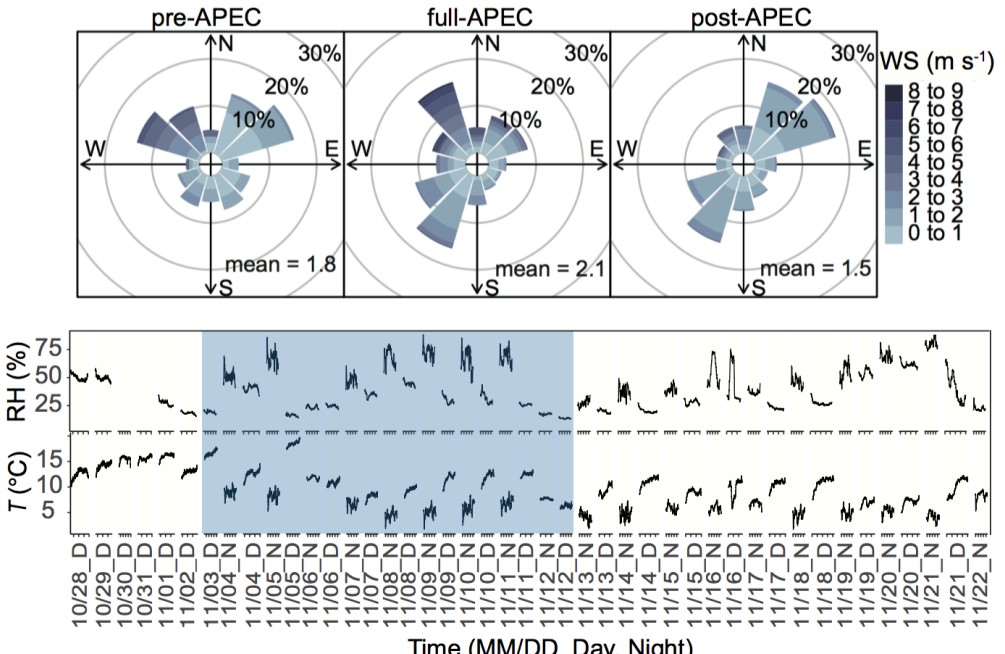

**Figure 2.** Upper panel: rose plots of wind direction (WD) and speed (WS) measured at the Nanjiao site during the pre-APEC (24 October
to 2 November, 2014), full-APEC (3–12 November, 2014), and post-APEC (12–22 November, 2014) periods. Lower panel: time series of
10   the on-road observed temperature (*T*) and relative humidity (RH) during the observation period; the blue shaded area indicates the full-
APEC period. APEC refers to the 2014 Asia-Pacific Economic Cooperation summit.





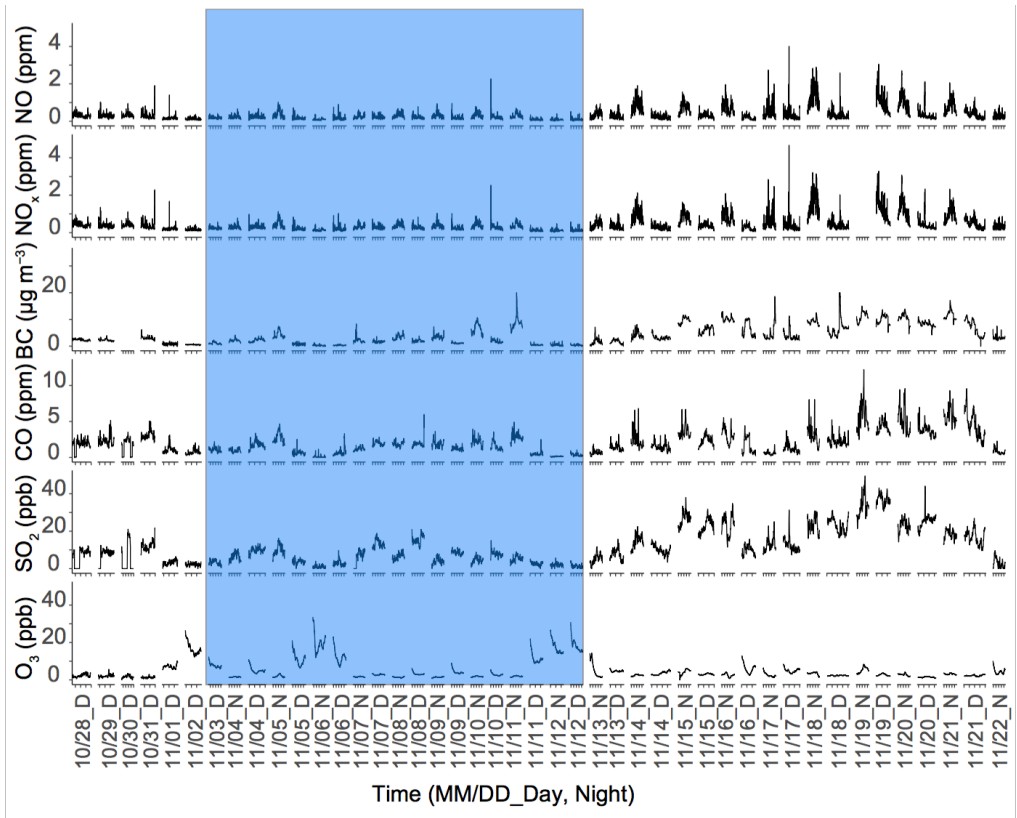

**Figure 3.** Time series of on-road mobile measurements of NO, $NO_x$, black carbon (BC), CO, $SO_2$, and $O_3$ concentrations around the 4th Ring Road of Beijing from 28 October to 22 November 2014. The blue shaded area indicates the full-APEC period.



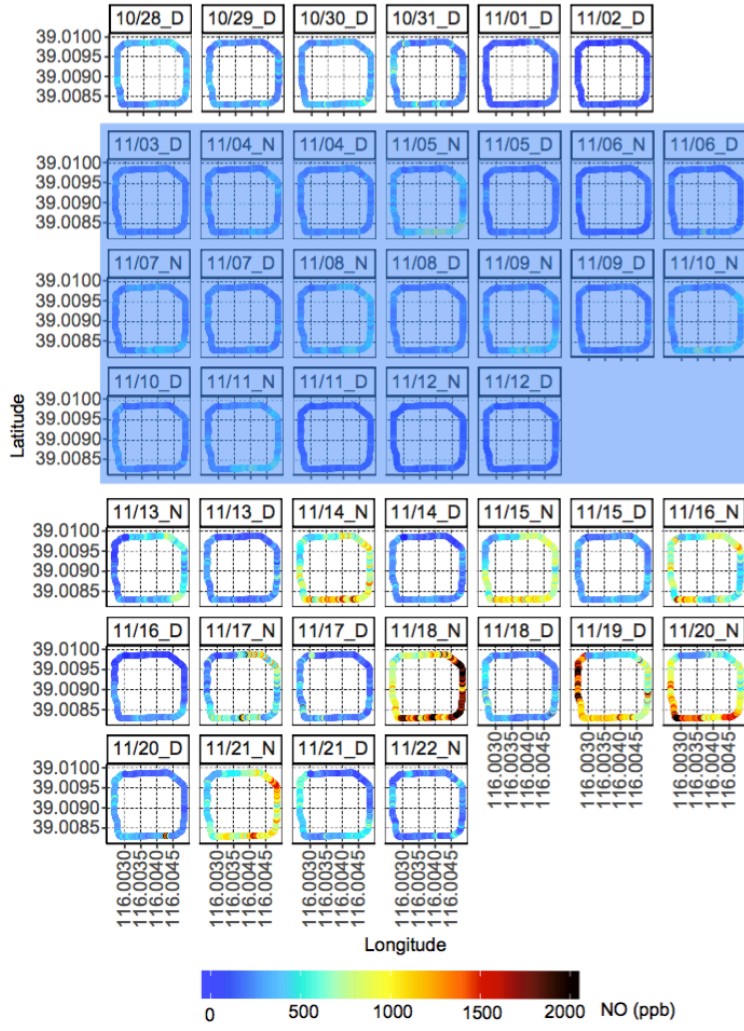

**Figure 4.** The spatial distributions of NO around the 4th Ring Road of Beijing from 28 October to 22 November 2014. The subtitle for each circuit indicates the observation time in the form "MM/DD_Day, Night". The blue shaded area indicates the full-APEC period.





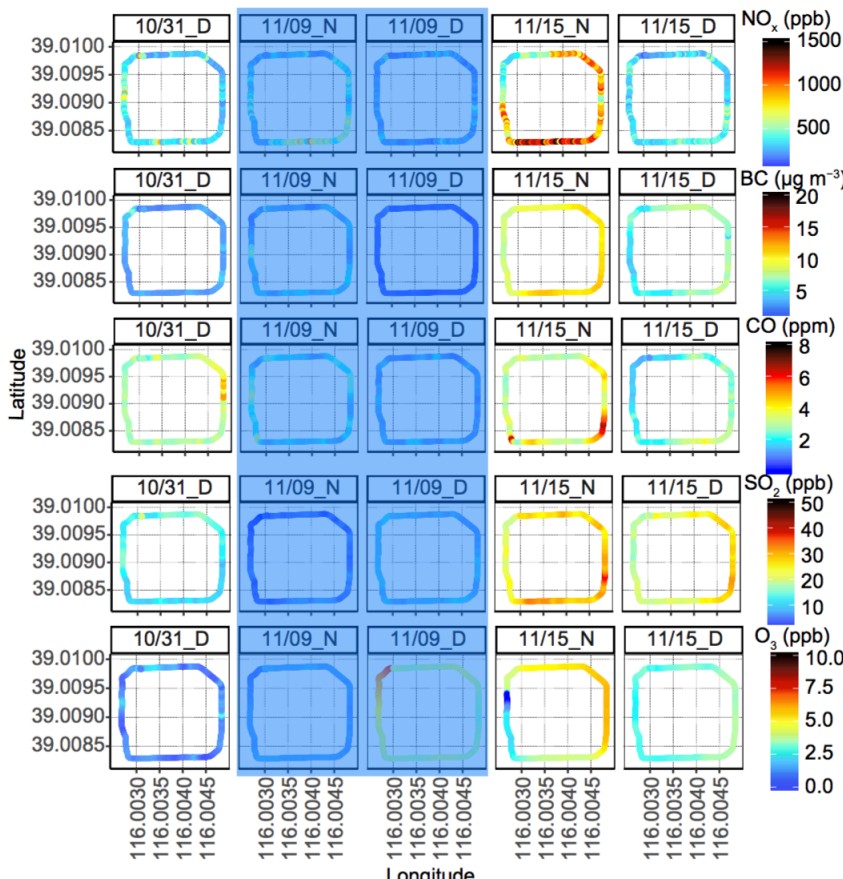

**Figure 5.** The spatial distributions of NO$_x$, BC, CO, SO$_2$, and O$_3$ observed around the 4th Ring Road of Beijing. 31 October, 9 November, and 15 November 2014 were selected to represent the pre-APEC, full-APEC, and post-APEC periods, respectively. The subtitle for each circuit indicates the observation time in the form "MM/DD_Day, Night". The blue shaded area indicates the full-APEC period.



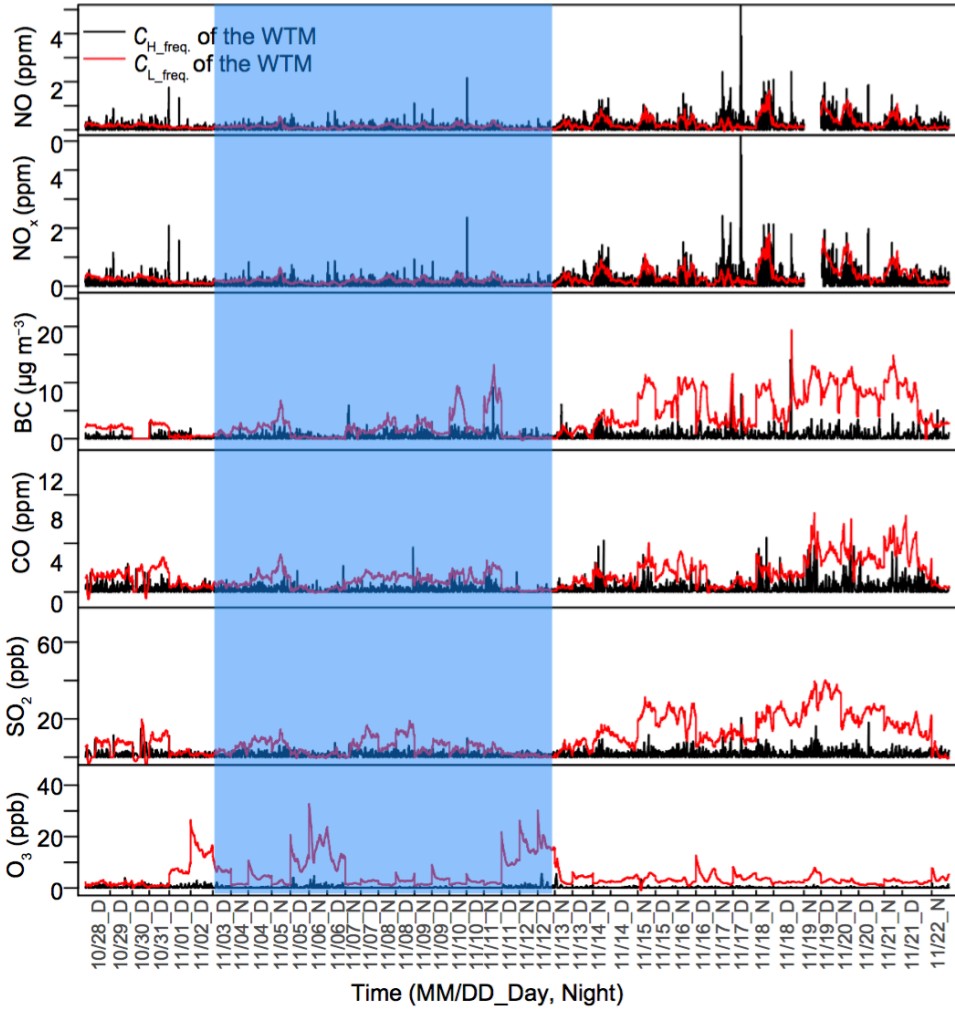

**Figure 6.** Time series of the decomposition results of the wavelet transform method (WTM) for NO, $NO_x$, BC, CO, $SO_2$, and $O_3$ (db6, eight levels; see text for details). The black lines represent the high-frequency components of the WTM, which should originate from immediate vehicle emissions. The red lines represent the low-frequency components of the WTM, which correspond to the atmospheric background plus accumulated on-road emissions. The blue shaded area indicates the full-APEC period.



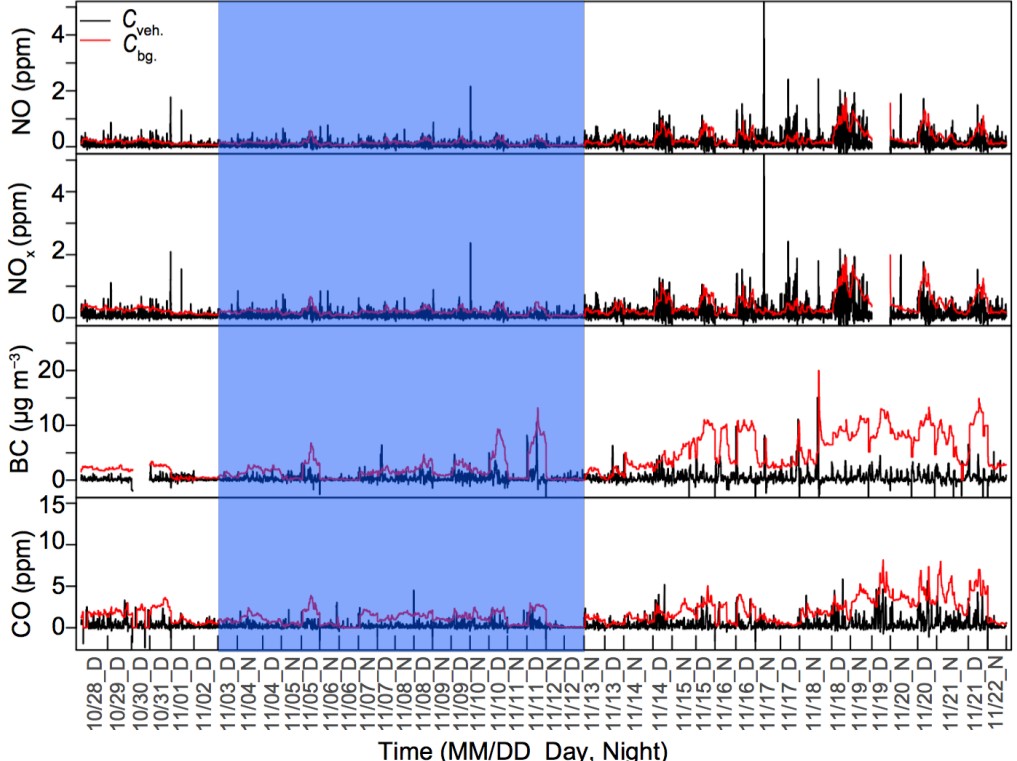

**Figure 7.** Time series of the separation results using the moving low percentile method (MLPM) for NO, $NO_x$, BC, CO, $SO_2$, and $O_3$. The red lines represent moving 5 min 5th percentiles of the measured pollutant concentrations to determine on-road background concentrations. The black lines represent estimated on-road emission concentrations, which were obtained by subtracting the background concentrations from the original concentrations. The blue shaded area indicates the full-APEC period.



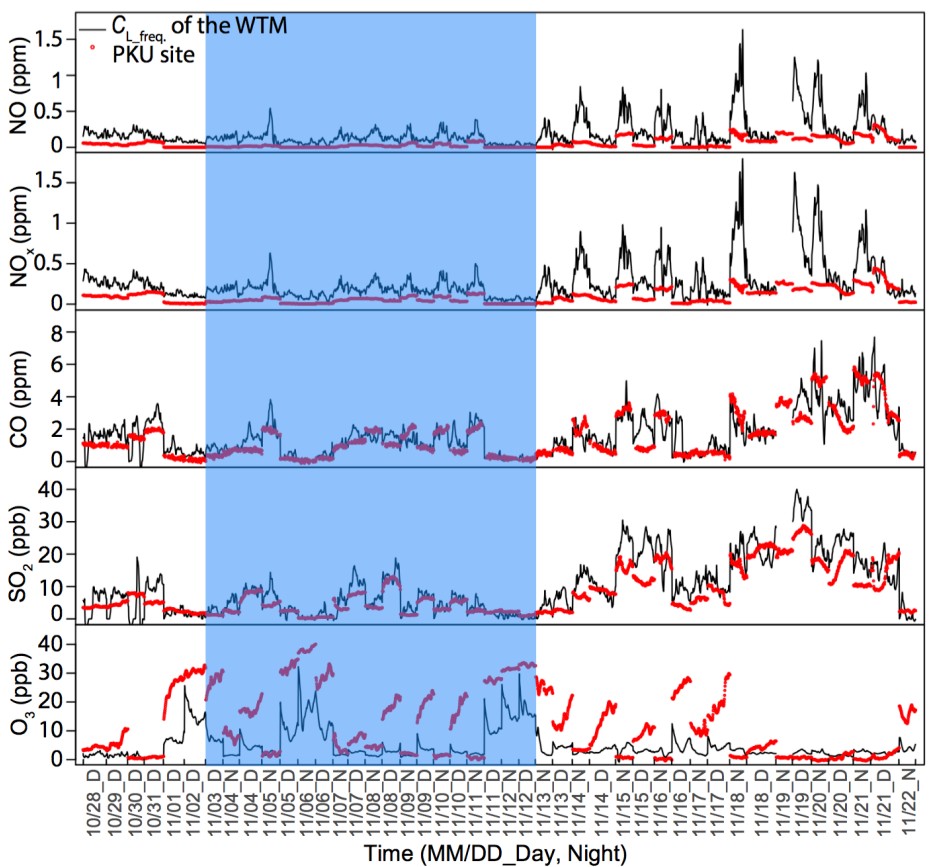

**Figure 8.** Comparisons between time series of the low-frequency components of the WTM and concentrations observed at the PKU site for NO, NO$_x$, BC, CO, SO$_2$, and O$_3$. The black lines represent the low-frequency components of the WTM. The red line represents the 1 min average concentrations of pollutants observed at the PKU site. The blue shaded area indicates the full-APEC period.

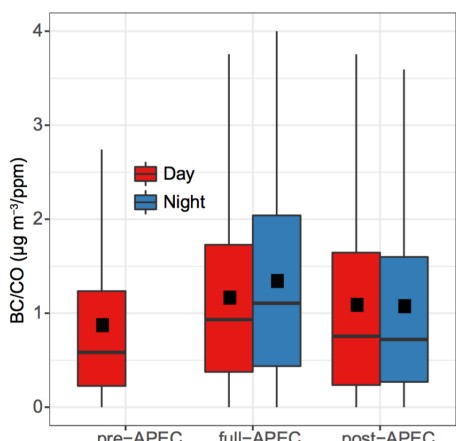

**Figure 9.** The distributions (boxplots) of BC/CO ratios derived from $C_{veh.}$ obtained from the WTM for the pre-, full-, and post-APEC periods. The black boxes and whiskers denote the 5th, 25th, 75th, and 95th percentiles. The black horizontal bars indicate median values. The black squares represent mean values. Red boxes indicate observations in the daytime and blue boxes indicate observations in the nighttime.



# Tables

Table 1. Air pollution control measures implemented during the pre-APEC, full-APEC, and post-APEC periods.

| Period | Control Measures | Details of Control Measures |
|---|---|---|
| Pre-APEC (28 October to 2 November 2014) | Forecasting and early warning | Forecast medium- and short-range air quality |
| | | Make air pollution emergency plan |
| | | Strengthen air quality monitoring and early warning forecasts |
| | Coal-emission controls | Reconstruction of coal-fired boilers to improve efficiency |
| | | Reduce output of coal-fired power plants |
| | | Postpone central heating |
| | Vehicle emission controls | Implement the normal licence plate restriction policy on weekdays (07:00–20:00) within (but not including) the 5th Ring Road |
| | | Local trucks and special vehicles forbidden from driving within (not including) the 5th Ring Road every day (06:00–23:00) |
| | | Yellow-labelled cars forbidden from driving within (including) the 6th Ring Road all day |
| | | Non-local trucks and special vehicles forbidden from driving within (not including) the 6th Ring Road every day (06:00–00:00) |
| | | Non-local vehicles with Beijing Entrance Permit forbidden from driving within (not including) the 6th Ring Road, passenger service vehicles forbidden from driving within (including) the 5th Ring Road on weekdays during peak hours (07:00–9:00 and 17:00–20:00) |
| | | Motorcycles forbidden from driving within (not including) the 6th Ring Road all day |
| Full-APEC (3–12 November 2014) | Vehicle emission controls | Implement odd–even licence plate rule every day (03:00–00:00) for the areas of Beijing, Tianjin, Hebei and parts of Shandong |
| | | Yellow-labelled cars, construction trucks, and hazardous material vehicles forbidden from driving in Beijing all day |
| | | Non-local trucks and special vehicles forbidden form driving within (including) the 6th Ring Road everyday (06:00–00:00) |



| Period | Measure type | Measures |
|---|---|---|
| | | Non-local vehicles forbidden from driving within (including) the 5th Ring Road on weekdays during peak hours (07:00–9:00 and 17:00–20:00) |
| | | Motorcycles forbidden from driving within (including) the 5th Ring Road all day |
| | | Stop use of government cars by 70%, and increase the capacity of public transport |
| | | Heavily-polluting trucks, tractors, and tricycles forbidden from driving within (including) the 6th Ring Road every day (06:00–00:00), and special postal trucks forbidden from driving within (not including) the 6th Ring Road every day (06:00–00:00) |
| | *Industrial emission controls* | Shut down 148 factories in Beijing |
| | | Reduce pollutant emissions by more than 30% |
| | | Reduce production or shut down more than 10,000 factories in Hebei, Tianjin, Shandong, Shanxi, Inner Mongolia, and other areas surrounding Beijing |
| | *Dust pollution controls* | Shut down or reduce the activities of dust emitting factories and outdoor construction in Beijing and its surrounding area |
| | | Increase road cleaning and spraying in Beijing |
| | *Emergency measures* | Further reduce production at state-owned enterprises |
| | | Reduce output of coal-fired units in Beijing by 40% |
| | | Strengthen the emission controls on special pollutants, such as VOCs in Beijing |
| | | National and public institutions, social organizations, and schools closed during 7–12 November |
| Post-APEC (13–22 November 2014) | *Vehicle emission controls* | Stop the odd–even licence plate rule and resume the normal tail number restriction policy |





**Table 2.** The mean concentrations of pollutants measured around the 4th Ring Road of Beijing and their relative changes between the pre-APEC, full-APEC, and post-APEC periods.

| Measured Species | All (mean ± SD) | Pre-APEC (mean ± SD) | Full-APEC (mean ± SD) | Post-APEC (mean ± SD) | D1[a] (%) | D2[b] (%) |
|---|---|---|---|---|---|---|
| NO (ppb) | 337.7 ± 330.4 | 242.4 ± 130 | 207.3 ± 135.7 | 488.2 ± 426.1 | −16.9 | −57.5 |
| $NO_x$ (ppb) | 416.6 ± 373.2 | 315.5 ± 155.6 | 262.7 ± 151.4 | 590.5 ± 478 | −20.1 | −55.5 |
| BC ($\mu g\ m^{-3}$) | 4.3 ± 3.7 | 1.6 ± 1.1 | 2.3 ± 2.4 | 6.7 ± 3.6 | +21.7 | −65.7 |
| CO (ppm) | 2.1 ± 1.6 | 1.8 ± 1.0 | 1.3 ± 0.9 | 2.8 ± 1.9 | −38.5 | −53.6 |
| $SO_2$ (ppb) | 11.8 ± 9.5 | 6.1 ± 5.0 | 6.1 ± 4.3 | 18.5 ± 9.5 | 0 | −67.0 |
| $O_3$ (ppb) | 5.0 ± 5.0 | 5.4 ± 5.8 | 6.8 ± 6.5 | 3.4 ± 1.7 | +20.6 | +100 |

SD: Standard Deviation

[a]D1 = (full-APEC − pre-APEC) ÷ pre-APEC

[b]D2 = (full-APEC − post-APEC) ÷ post-APEC

**Table 3.** The on-road emission concentrations obtained from the WTM for NO, $NO_x$, BC, CO, $SO_2$, and $O_3$ with different mother wavelet functions (i.e., db4–db8) and decomposition levels (5–9 levels).

| | NO (ppb) (mean ± SD) | $NO_x$ (ppb) (mean ± SD) | BC ($\mu g\ m^{-3}$) (mean ± SD) | CO (ppm) (mean ± SD) | $SO_2$ (ppb) (mean ± SD) | $O_3$ (ppb) (mean ± SD) |
|---|---|---|---|---|---|---|
| db4_level5 | 140.4 ± 190.2 | 148.0 ± 197.0 | 0.5 ± 0.6 | 0.2 ± 0.3 | 0.8 ± 0.6 | 0.1 ± 0.1 |
| db4_level6 | 140.7 ± 188.9 | 150.7 ± 197.5 | 0.5 ± 0.7 | 0.3 ± 0.4 | 1.3 ± 1.2 | 0.2 ± 0.2 |
| db4_level7 | 142.0 ± 196.0 | 152.1 ± 204.8 | 0.5 ± 0.8 | 0.4 ± 0.5 | 1.6 ± 1.6 | 0.2 ± 0.4 |
| db4_level8 | 138.9 ± 199.3 | 149.1 ± 208.2 | 0.6 ± 0.9 | 0.4 ± 0.6 | 1.7 ± 1.8 | 0.3 ± 0.5 |
| db4_level9 | 138.6 ± 201.3 | 148.7 ± 210.8 | 0.6 ± 1.0 | 0.4 ± 0.6 | 1.7 ± 1.9 | 0.3 ± 0.5 |
| db5_level8 | 140.0 ± 201.4 | 150.3 ± 210.9 | 0.6 ± 0.9 | 0.4 ± 0.6 | 1.8 ± 1.9 | 0.3 ± 0.5 |
| db6_level8 | 140.3 ± 200.6 | 150.3 ± 210.0 | 0.6 ± 0.9 | 0.4 ± 0.6 | 1.8 ± 1.9 | 0.3 ± 0.5 |
| db7_level8 | 139.3 ± 198.6 | 149.4 ± 207.6 | 0.6 ± 0.9 | 0.4 ± 0.6 | 1.7 ± 1.8 | 0.3 ± 0.5 |
| db8_level8 | 138.7 ± 199.3 | 148.9 ± 208.5 | 0.6 ± 0.9 | 0.4 ± 0.6 | 1.7 ± 1.8 | 0.3 ± 0.5 |
| Uncertainty[a] (%) | 0.3 | 0.3 | 1.1 | 3.2 | 2.6 | 1.4 |

[a]Uncertainty = the SD of on-road concentrations estimated using different schemes in the WTM divided by the mean pollutant concentration.





**Table 4.** The mean on-road emission concentrations estimated using the moving low percentile method (MLPM) for NO, NO$_x$, BC, CO, SO$_2$, and O$_3$ with different time windows (i.e., 1–10 min) and percentiles (i.e., 1%, 5% and 10%).

| | NO (ppb) (mean ± SD) | NO$_x$ (ppb) (mean ± SD) | BC (µg m$^{-3}$) (mean ± SD) | CO (ppm) (mean ± SD) | SO$_2$ (ppb) (mean ± SD) | O$_3$ (ppb) (mean ± SD) |
|---|---|---|---|---|---|---|
| 5%_1min | 128.5 ± 194.3 | 136.8 ± 202.4 | 0.6 ± 0.9 | 0.4 ± 0.6 | 1.7 ± 1.9 | 0.4 ± 0.8 |
| 5%_3min | 119.3 ± 195.8 | 126.5 ± 204.2 | 0.5 ± 1.0 | 0.4 ± 0.6 | 1.5 ± 1.9 | 0.2 ± 0.5 |
| 5%_5min | 122.7 ± 198.8 | 130.0 ± 207.3 | 0.5 ± 1.0 | 0.4 ± 0.6 | 1.6 ± 2.0 | 0.3 ± 0.5 |
| 5%_8min | 135.4 ± 210.7 | 144.2 ± 221.8 | 0.7 ± 1.3 | 0.5 ± 0.7 | 2.1 ± 2.4 | 0.3 ± 0.6 |
| 5%_10min | 142.7 ± 216.2 | 152.6 ± 228.0 | 0.8 ± 1.4 | 0.5 ± 0.7 | 2.3 ± 2.5 | 0.4 ± 0.7 |
| 1%_5min | 138.1 ± 204.1 | 147.8 ± 214.0 | 0.5 ± 1.0 | 0.4 ± 0.6 | 1.7 ± 2.0 | 0.3 ± 0.5 |
| 10%_5min | 107.2 ± 193.2 | 113.8 ± 202.1 | 0.5 ± 1.0 | 0.3 ± 0.6 | 1.4 ± 2.0 | 0.2 ± 0.5 |
| Uncertainty[a] (%) | 3.4 | 3.0 | 2.6 | 3.0 | 2.6 | 1.5 |

[a]Uncertainty = the SD of on-road concentrations estimated using different schemes in the MLPM divided by the mean pollutant concentration.

**Table 5.** The proportion of on-road emission concentrations decomposed by the WTM for the entire observation period (All) and the pre-APEC, full-APEC, and post-APEC periods.

| Measured Species | All (%) (mean ± SD) | pre-APEC (%) (mean ± SD) | full-APEC (%) (mean ± SD) | post-APEC (%) (mean ± SD) |
|---|---|---|---|---|
| NO | 42 ± 51 | 38 ± 63 | 41 ± 56 | 42 ± 51 |
| NO$_x$ | 36 ± 47 | 32 ± 56 | 35 ± 52 | 37 ± 47 |
| BC | 14 ± 20 | 17 ± 29 | 22 ± 22 | 11 ± 24 |
| CO | 19 ± 32 | 22 ± 36 | 21 ± 27 | 18 ± 30 |
| SO$_2$ | 14 ± 17 | 26 ± 30 | 23 ± 23 | 11 ± 19 |
| O$_3$ | 6 ± 9 | 11 ± 10 | 4 ± 7 | 6 ± 20 |





**Table 6.** The mean on-road emission concentrations obtained from the WTM and their relative changes between the pre-APEC, full-APEC, and post-APEC periods.

| db6_level8 | Pre-APEC (mean ± SD) | Full-APEC (mean ± SD) | Post-APEC (mean ± SD) | D1[a] (%) | D2[b] (%) |
|---|---|---|---|---|---|
| NO (ppb) | 92.5 ± 104.8 | 90.4 ± 96.6 | 206.1 ± 266 | −2.3 | −56.1 |
| NO$_x$ (ppb) | 101.5 ± 115.3 | 97.5 ± 102.6 | 218.7 ± 277.6 | −3.9 | −55.4 |
| BC ($\mu$g m$^{-3}$) | 0.3 ± 0.4 | 0.5 ± 0.7 | 0.7 ± 1.1 | +66.7 | −28.6 |
| CO (ppm) | 0.4 ± 0.5 | 0.3 ± 0.3 | 0.5 ± 0.7 | −25.0 | −40.0 |
| SO$_2$ (ppb) | 1.6 ± 2 | 1.4 ± 1.3 | 2 ± 2.1 | −12.5 | −30.0 |
| O$_3$ (ppb) | 0.6 ± 0.6 | 0.2 ± 0.4 | 0.2 ± 0.4 | −66.7 | 0.0 |

[a]D1 = (full-APEC − pre-APEC) ÷ pre-APEC

[b]D2 = (full-APEC − post-APEC) ÷ post-APEC

**Table 7.** The mean on-road emission concentrations obtained from WTM and their relative changes between the pre-APEC, full-APEC, and post-APEC periods, divided into daytime and nighttime.

| | Measured Species | Pre-APEC (mean ± SD) | Full-APEC (mean ± SD) | Post-APEC (mean ± SD) | D1[a] (%) | D2[b] (%) |
|---|---|---|---|---|---|---|
| Day | NO (ppb) | 92.5 ± 104.8 | 74.6 ± 89.9 | 149.2 ± 237.4 | −19.4 | −50.0 |
| | NO$_x$ (ppb) | 101.5 ± 115.3 | 83.5 ± 98.1 | 158.3 ± 243.6 | −17.7 | −47.3 |
| | BC ($\mu$g m$^{-3}$) | 0.3 ± 0.4 | 0.3 ± 0.3 | 0.8 ± 1.3 | 0.0 | −62.5 |
| | CO (ppm) | 0.4 ± 0.5 | 0.2 ± 0.3 | 0.4 ± 0.5 | −50.0 | −50.0 |
| | SO$_2$ (ppb) | 1.6 ± 2.0 | 1.3 ± 1.2 | 1.7 ± 1.8 | −18.8 | −23.5 |
| | O$_3$ (ppb) | 0.6 ± 0.6 | 0.3 ± 0.6 | 0.2 ± 0.2 | −50.0 | +50.0 |
| Night | NO (ppb) | - | 98.5 ± 97.0 | 283.8 ± 282.9 | - | −65.3 |
| | NO$_x$ (ppb) | - | 104.3 ± 100.9 | 301.2 ± 299.1 | - | −65.4 |
| | BC ($\mu$g m$^{-3}$) | - | 0.6 ± 0.9 | 0.7 ± 0.9 | - | −14.3 |
| | CO (ppm) | - | 0.3 ± 0.3 | 0.6 ± 0.9 | - | −50.0 |
| | SO$_2$ (ppb) | - | 1.4 ± 1.3 | 2.3 ± 2.4 | - | −39.1 |
| | O$_3$ (ppb) | - | 0.2 ± 0.5 | 0.2 ± 0.5 | - | 0.0 |

[a]D1 = (full-APEC − pre-APEC) ÷ pre-APEC

[b]D2 = (full-APEC − post-APEC) ÷ post-APEC