# Peer review of "Using wavelet transform to analyse on-road mobile measurements of air pollutants: a case study to evaluate vehicle emission control policies during the 2014 APEC summit"

_Atmospheric Chemistry and Physics, 2019_

## Referee Comment (RC1) · Anonymous Referee #2 · 10 Jun 2019

This work evaluates the control effect for vehicle emissions during the 2014 APEC summit by using on-road mobile measurements in combination with a newly developed wavelet transform method (WTM). The WTM successfully distinguishes the signals of air pollutant emitted by vehicle exhausts from those existing in ambient air and thus provides a good reference for similar data analysis. The results confirm the control policies on vehicle emissions were effective. The manuscript is generally well written with clear logic and full discussion. It is recommended to be accepted after addressing the following minor comments.

1. Line 15 on Page 3, sampling inlet is very important for the data quality in an on-road mobile measurements. Elaborate the detailed configuration of the inlet including material, size, shape, direction, and the retention time of the air flow.

2. Line 40 on Page 6 and Line 8 and 18 on Page 7, what are the major reasons for the elevated values of NOx, BC and ozone in the southern part of the Ring Road, busy traffic, concentrated industries, or both?

3. Line 4 on Page 7, point out the criteria of the selected three days, for example similar meteorological conditions and/or pollution levels.

4. Line28-31 on Page 9, in addition to the missing nighttime data and the gradually implemented control policies in the pre-APEC period, do the less favorable meteorological conditions (see Line 5 on Page 6) and the increased energy consumption due to household heating also contribute to the higher decrease in Cveh relative to the post-APEC period?

———————————————————————

---

## Referee Comment (RC2) · Anonymous Referee #1 · 24 Jul 2019

General comments:

This manuscript shows the observed concentration of the air pollutions in Beijing before/after the air quality control by the government for the 2014 APEC summit. The manuscript is generally well written, but several points should be clarified. First, although the WTM-observed results are evaluated using other instruments, i.e., moving low percentile method (MLPM), the evaluation of the MLPM itself is unclear. Second, the quantification to evaluate the results is inadequate in some parts like Figures 7 and 8. Third, O3 is the secondary component, so its discussion should be separated

from that in the primary components, i.e., NO, (NOx), BC, CO and SO2. In overall, the manuscript would be acceptable for publication if these comments can be satisfactorily addressed.

Specific comments:

P.1, L27-28: I don't think the current manuscript adequately shows the feasibility and stability of the WTM. Also, which point of the WTM is useful for the evaluation of pollution control policies? These are unclear.

P.5, L24-25: What is the moving low percentile method (MLPM)?

P.5, L33-L36: Do the authors discuss an impact of the assumption on main results?

P.6, L17-L19: The authors use "comparable", but this is not suitable for CO and SO2 shown in L20. The difference in CO and SO2 between this study and the reference is more than twice. Also, I think the level of O3 in China is relatively lower than that in the other regions.

P.8, L12-L13: How about SO2 and O3?

P.8, L30-L31: There is no information about the statistic metrics such as bias, correlation coefficient, and root-mean-square-error. Please show them for the comparison. In addition, the uncertainties are estimated in Tables 3 and 4, but actually how are they calculated? I don't exactly understand the meaning of "the SD of on-road concentrations estimated using different schemes in the WTM divided by the mean pollutant concentration" on Tables 2 and 3. What is the different scheme? Does the SD determine the uncertainty?

Table 5: Does it show the ratio of CH to the sum of CH and CL? If so, this result may show a failure of the decomposition of CH and CL. I expected the value in full-APEC is lower than that in the other period, but it is not. Perhaps, the total amount of the sum of CH and CL in full-APEC is also lower than that in the other period, so that the proportion in full-APEC is not lowest among the periods. In this sense, I don't understand what

the main message in this table is.

Figure S4: Are the R values corrected? They look high, even though the points are well scattered. Please check them again. The authors can insert the statistic metrics into Figures 7 and 8.

Technical comments:

P.1, L23-L24 & P.10, L16-L18: 50% → 50.0%

P.3, L4: "In Sect. 3" can be removed.

P.3, L31: Where is the 5th Ring in Figure 1?

P.6, L11-13: These references may be not exactly related to the current manuscript. Are these important in this manuscript?

P.13, L25: Please add the name of journal.

Figures 4-8: The blue shaded covers the important color. The shaded is not good.

Figure 7: How about SO2 and O3?

Figure 8: How about BC?

Table 6: Does it show the sum of CH and CL?

Table 7: Does it show the CH?

---

## Author Comment (AC1) · 4 Sep 2019

**Response to the Comments of Referees**

**Using wavelet transform to analyse on-road mobile measurements of air pollutants: a case study to evaluate vehicle emission control policies during the 2014 APEC summit**

**Yingruo Li, Ziqiang Tan, Chunxiang Ye, Junxia Wang, Yanwen Wang, Yi Zhu, Pengfei Liang, Xi Chen, Yanhua Fang, Yiqun Han, Qi Wang, Di He, Yao Wang, and Tong Zhu**

We thank the referees for the critical comments, which are very helpful in improving the quality of the manuscript. We have made revision based on the critical comments and suggestions of the reviewers. Our point-by-point responses to the comments are listed in the following.

**Referee #1:**

**Comment NO.1:** *This manuscript shows the observed concentration of the air pollutions in Beijing before/after the air quality control by the government for the 2014 APEC summit. The manuscript is generally well written, but several points should be clarified. First, although the WTM-observed results are evaluated using other instruments, i.e., moving low percentile method (MLPM), the evaluation of the MLPM itself is unclear. Second, the quantification to evaluate the results is inadequate in some parts like Figures 7and 8. Third, $O_3$ is the secondary component, so its discussion should be separated from that in the primary components, i.e., NO, (NOx), BC, CO and SO2. In overall, the manuscript would be acceptable for publication if these comments can be satisfactorily addressed.*

**Response:** We agree with the comments of the referee and made revision according to the referee's advice. The referee's concerns are responded and addressed as bellow.

**(1). The comparison of the WTM and the MLPM.**

It is true there is no absolute value for both the MLPM (moving low percentile method) and the WTM to compare with. The MLPM is developed by Bukowiecki et al. (2002) and is a tested method for separating concentration of on-road background (i.e. $C_{bg.}$ or $C_{L\_freq.}$) and concentration from immediate vehicle emissions (i.e. $C_{veh.}$ or $C_{H\_freq.}$), which is useful in evaluation of on-road emission reduction. In this study, we developed the WTM method and compared it with the MLPM. The result suggests that both the WTM and the MLPM are capable of effectively separating $C_{veh.}$ and thus evaluating on-road emission reduction during the 2014 APEC summit. This point has been further clarified in the revised manuscript (Page 5 Line 29-31, Page 8 Line 34-40, Page 9 Line 15-18).

**(2). The quantification in Figure 7 and 8.**

We agree that the quantification in Figure 7 and 8 is essential for the evaluation of results of the WTM. Therefore we strengthened this part in our revised manuscript following the referee's comments. The correlation coefficient ($R^2$) of $C_{veh.}$ decomposed by the WTM and the MLPM was calculated and showed in Fig.S4. The correlation of

the WTM and the MLPM in on-road emission concentration ($C_{veh.}$) is high. The correlation coefficients ($R^2$) between $C_{veh.}$ from the WTM and the MLPM were 0.996, 0.994, 0.894, and 0.840 for NO, $NO_x$, CO, and BC, respectively (Fig. S4). This comparison indicates the agreement of both methods in signal decomposition.

[Figure]

**(3). The discussions on $O_3$.**

We agree that $O_3$ is not a tracer for vehicle emissions. As a measured pollutant in our experiment, it was presented in the general discussion section. However, as we discuss the on-road emission reduction, $O_3$ is not included as suggested. Related revision in our manuscript can be found Page 8 Line 38-40.

**Changes in Manuscript:** We made revision according to the referee's advice, please refer to Page 5 Line 29-31, Page 8 Line 34-40, Page 9 Line 15-18 in the revised manuscript and Fig. S4 and S5 in the revised supplement for the revisions.

**Comment NO.2:** *P.1, L27-28: I don't think the current manuscript adequately shows the feasibility and stability of the WTM. Also, which point of the WTM is useful for the evaluation of pollution control policies? These are unclear.*

**Response:** We agree that the sentence here is arbitrary and unclear. We rewrote the sentences in the revised manuscript. "Using on-road mobile measurements in combination with the WTM we developed a newly method for the evaluation of pollution control policies."

**Changes in Manuscript:** We rewrote the related sentence, please refer to Page 1 Line 27-28 in the revised manuscript for this revision.

**Comment NO.3:** *P.5, L24-25: What is the moving low percentile method (MLPM)?*

**Response:** The moving low percentile method (MLPM) is developed by Bukowiecki

et al. (2002) and is a tested method for separating concentration of on-road background (i.e. $C_{bg.}$ or $C_{L\_freq.}$) and concentration from immediate vehicle emissions (i.e. $C_{veh.}$ or $C_{H\_freq.}$), which is useful in evaluation of on-road emission reduction.

**Changes in Manuscript:** We added the definition of the MLPM in the revised manuscript, please refer to Page 5 Line 29-31 in the revised manuscript for the changes.

**Comment NO.4:** *P.5, L33-L36: Do the authors discuss an impact of the assumption on main results?*

**Response:** Yes, we added discussion on the impact of the assumption. The referee's concern is responded below.

Firstly, we have considered the impact of meteorological conditions in the experimental design. All observation trips started at the same period of 1 day (i.e. about 10:00 am LT in the daytime and about 1:00 am LT in the nighttime), with relatively stable atmospheric boundary layer during the selected observation period, which can eliminate the meteorological influence to some extent. We chose the wide main road (i.e. the 4th Ring Road) for observation to reduce the impact of micro-scale turbulence and meteorological disturbances which is complex on narrow roads. Secondly, direct measurements of wind speed and wind direction from the mobile lab during driving may have large deviations (Johansson, 2009; Wang et al., 2011). The observation data of wind speed and direction at the ground based site (such as the Nanjiao site) is hourly mean data, with low time resolution and limited spatial representation. It is difficult to obtain high time-resolution and accurate meteorological data matching the concentration of gaseous pollutants. The lack of high time resolution meteorological data makes it difficult to accurately evaluate and eliminate the meteorological impacts. Therefore, we thought the short-term micro-scale meteorology has little effect on our assessment results, and the discussion and accurate quantification of micro-scale meteorological influences are beyond the scope of this study. In this paper, we assumed that the high-frequency component is confined to much shorter time scales than those of the meteorological factors, and that changes in meteorological conditions were reflected only in the low-frequency component, such that changes in meteorological conditions did not influence our evaluation of control polices.

**Changes in Manuscript:** We added discussion on the impact of the assumption according to the referee's concerns in the revised manuscript, please refer to Page 6 Line 16-29 for the changes.

**Comment NO.5:** *P.6, L17-L19: The authors use "comparable", but this is not suitable for CO and $SO_2$ shown in L20. The difference in CO and $SO_2$ between this study and the reference is more than twice. Also, I think the level of $O_3$ in China is relatively lower than that in the other regions.*

**Response:** Accepted. The word "comparable" used here is not rigorous. Our results indeed showed that the air pollution problems in Beijing and among the worst in the world (Zhu et al., 2016). The differences in concentration of CO and $SO_2$ in this study with the reference (Zhu et al., 2016) may ascribed to the different measurement area and seasons. As for the low level of $O_3$ concentration in this study mainly due to the high on-road concentration of NO induced strong titration reaction of $O_3$. The ground based measurement showed that the $O_3$ pollution in Beijing is quite severe and the

hourly average concentration of $O_3$ in the downwind area of Beijing ever reached 286 ppb (Wang et al., 2008).

**Changes in Manuscript:** We have adjusted the related sentences in the revised manuscript to make more rigorous discussion. Please refer to Page 6 Line 33-34 and Line 42-43 for the changes.

**Comment NO.6:** *P.8, L12-L13: How about $SO_2$ and $O_3$?*

**Response:** Vehicle emissions have quite limited contribution to the concentration of $SO_2$ (Zhu et al., 2016). The spatial distribution of $SO_2$ also indicates that it is not a typical tracer for vehicle emissions (Fig. 5). $O_3$ is a secondary pollutant which is affected by the titration of high-concentration NO and regional transport. The atmospheric physical and chemical processes of $O_3$ are complicated, and it is not an ideal tracer for studying vehicle emissions and evaluating vehicle control policies. Therefore, we did not focus on $SO_2$ and $O_3$ during the evaluation of vehicle control policies in this study.

**Changes in Manuscript:** We gave the clarification for the reason of missing discussion on $SO_2$ and $O_3$ in Sect. 3. Please refer to Page 8 Line 38-40 for the changes.

**Comment NO.7:** *P.8, L30-L31: There is no information about the statistic metrics such as bias, correlation coefficient, and root-mean-square-error. Please show them for the comparison. In addition, the uncertainties are estimated in Tables 3 and 4, but actually how are they calculated? I don't exactly understand the meaning of "the SD of on-road concentrations estimated using different schemes in the WTM divided by the mean pollutant concentration" on Tables 2 and 3. What is the different scheme? Does the SD determine the uncertainty?*

**Response:** We agree that using the "the SD of on-road concentrations estimated using different schemes in the WTM divided by the mean pollutant concentration" to calculate the uncertainty is unreasonable here and information of statistic metrics in needed. We made revision to address these issues according to the referee's advice. The different scheme in Table 2 and 3 is the decomposition schemes with different decomposition parameters, such as different mother wavelet function ('dbN') and decomposition levels (level4-8) for the WTM as well as different percentiles (1%, 5%, and 10%) and time windows (1 min, 3min, 5 min, 8min and 10 min) for the MLPM. In the revised manuscript we use the SD to evaluate the stability of the method. The SD of the WTM results is smaller than that of the MLPM method, which indicated the WTM is a more stable method for decomposition. The correlation coefficient ($R^2$) of $C_{veh.}$ decomposed by the WTM and the MLPM was calculated and showed in Fig.S4.

**Changes in Manuscript:** We made changes in Table 2 and Table 3 as well as the discussions in the revised manuscript to give more clear and reasonable analysis according to the referee's suggestions. Please refer to Page 8 Line 34-40, Page 9 Line 15-18, Page 24 Table 3, and Page 25 Table 4 in the revised manuscript and Page 4 Figure S4 in the revised supplement for the revisions.

**Comment NO.8:** *Table 5: Does it show the ratio of $C_H$ to the sum of $C_H$ and $C_L$? If so, this result may show a failure of the decomposition of $C_H$ and $C_L$. I expected the value in full-APEC is lower than that in the other period, but it is not. Perhaps, the total*

*amount of the sum of $C_H$ and $C_L$ in full-APEC is also lower than that in the other period, so that the proportion in full-APEC is not lowest among the periods. In this sense, I don't understand what the main message in this table is.*

**Response:** The total amount of the sum of $C_H$ and $C_L$ in full-APEC is also lower than that in the other period, so that the proportion in full-APEC is not lowest among the periods. But the proportions analysis here can give some clue of the reasonable of the WTM results and also further indicate that NO and $NO_x$ is the most suitable and typical tracers for the evaluation of vehicle emissions.

**Comment NO.9:** *Are the R values corrected? They look high, even though the points are well scattered. Please check them again. The authors can insert the statistic metrics into Figures 7 and 8.*

**Response:** Accepted. We have carefully checked the results and the *R* values and corrected the errors in the revised manuscript. The correlation coefficients ($R^2$) between $C_{L\_freq.}$ from the WTM and PKU site observations were 0.748, 0.715, 0.912, 0.927, and 0.752 for NO, $NO_x$, CO, $SO_2$, and $O_3$, respectively (Fig. S5). We tried insert the statistic metrics into Figure 7 and 8, but the figures are not good for reading. So we keep them in the supplement.

**Changes in Manuscript:** Please refer to Page 9 Line 24 in the revised manuscipt and Page 5 Fig.S5 in the revised supplement for the changes.

[Figure]

**Comment NO.10:** *P.1, L23-L24 & P.10, L16-L18: 50% →50.0%; P.3, L4: "In Sect. 3" can be removed.*

**Response:** Accepted.

**Changes in Manuscript:** We have check the decimal places be quoted of the results and removed "In Sect. 3". Please refer to Page 1 Line 24-26, Page 10 Line 23-24, Page 11 Line 5-8 and Page 3 Line 4 in the revised manuscript for the revision.

**Comment NO.11:** *P.3, L31: Where is the 5th Ring in Figure 1?*

**Response:** The 5th Ring Road of Beijing was marked by the green line in the following Figure.

[Figure]

**Changes in Manuscript:** We revised Figure 1 in the revised Manuscript to give the information of the 5th Ring Road for easy reading. Please refer to Page 15 Figure 1 in the revised manuscript for the revision.

**Comment NO.12:** *P.6, L11-13: These references may be not exactly related to the current manuscript. Are these important in this manuscript?*

**Response:** We agree that the references and discussion here is beyond the scope of this paper.

**Changes in Manuscript:** We deleted the sentences and corresponding references in the revised manuscript, please refer to Page 6 Line 14-15 in the revised manuscript for the changes.

**Comment NO.13:** P.13, L25 *Please add the name of journal.*

**Response:** Accepted. We have removed the unrelated discussions and sentences in the revised manuscript as mentioned in Comment NO. 12. We also check carefully for the format of the references.

**Changes in Manuscript:** Please refer to Page 6 and Page 14 in the revised manuscript for the revisions.

**Comment NO.14:** *Figures 4-8: The blue shaded covers the important color. The shaded is not good.*

**Response:** We agree that the blue shaded covers the important color in Figure 4-8 and is not good for reading.

**Changes in Manuscript:** We have changed the way of marking for APEC period. Please refer Figure 4-8 in the revised manuscript for the revisions.

**Comment NO.15:** *Figure 7: How about SO$_2$ and O$_3$?*

**Response:** As we mentioned in Comment NO.6. SO$_2$ and O$_3$ is not typical tracer for the evaluation and investigation of vehicle emissions. Thus, we do not give the composition results for SO$_2$ and O$_3$ in Figure7.

**Comment NO.16:** *Figure 8: How about BC?*

**Response:** Due to lack of BC observation data at the PKU site, there is no comparison result for BC in Figure 8.

**Comment NO.17:** *Table 6: Does it show the sum of CH and CL?*

**Response:** Table 6 showed the mean on-road emission concentrations, which is expressed as $C_{veh.}$ or $C_{H\_freq.}$ in the manuscript.

**Changes in Manuscript:** This point has been clarified in the revised manuscript. Please refer Page 26 Line 1 for the changes.

**Comment NO.18:** *Table 7: Does it show the CH?*

**Response:** Table 7 showed the mean on-road emission concentrations, which is expressed as $C_{veh.}$ or $C_{H\_freq.}$ in the manuscript.

**Changes in Manuscript:** This point has been clarified in the revised manuscript. Please refer Page 26 Line 6 for the changes.

**Referee #2:**

**Comment NO.1:** *This work evaluates the control effect for vehicle emissions during the 2014 APEC summit by using on-road mobile measurements in combination with a newly developed wavelet transform method (WTM). The WTM successfully distinguishes the signals of air pollutant emitted by vehicle exhausts from those existing in ambient air and thus provides a good reference for similar data analysis. The results confirm the control policies on vehicle emissions were effective. The manuscript is generally well written with clear logic and full discussion. It is recommended to be accepted after addressing the following minor comments. 1. Line 15 on Page 3, sampling inlet is very important for the data quality in an on road mobile measurements. Elaborate the detailed configuration of the inlet including material, size, shape, direction, and the retention time of the air flow.*

**Response:** Accepted. The sampling inlet is important for the data quality of on-road mobile measurements. The sampling system included inlets for gaseous pollutants, volatile organic compounds (VOCs), particulate matter and black carbon. The inlets for gaseous pollutants and particulate matter are located separately at the front top of the van, 3.2 m above the ground to reduce sampling of the van exhaust. The gaseous sampling inlet is composed of a glass manifold and Teflon branched tubes. The total length of the glass manifold is about 2 m, and the retention time between the sampling inlet and the instrument is about 8 s. Detailed configuration information of the inlet can be found in previous study (Wang et al., 2009).

**Changes in the manuscript:** We have added the configuration information of the inlet system to the revised manuscript. Please refer to the revised manuscript from Page 3 Line 15-20.

**Comment NO.2:** *Line 40 on Page 6 and Line 8 and 18 on Page 7, what are the major reasons for the elevated values of NO$_x$, BC and ozone in the southern part of the Ring Road, busy traffic, concentrated industries, or both?*

**Response:** The distribution of NO$_x$, BC and O$_3$ on the 4th Ring Road of Beijing was determined both by vehicle emissions and regional transport (Wang et al., 2009; Zhu et al., 2016). The previous studies indicated that vehicle emission have important contribution to the NO$_x$ and BC concentration (Cheng et al., 2013; Westerdahl et al., 2009). It is well known that the southern part of Beijing is close to the economically developed areas, with busier traffic and relatively dense industrial emissions in the southern surrounding area of Beijing. Thus, it is speculated that the major reason for the elevated values of NO$_x$ and BC in the southern part of the 4th Ring Road is busy traffic, while the concentrated industries also is one of the reasons. The high ozone values in the southern part is mainly due to the regional transport from southern area where ozone photochemical reaction is strong (Li et al., 2016).

**Changes in Manuscript:** We added the discussion on the reasons of the distributions of the gaseous pollutants in the revised manuscript, please refer to Page 7 Line 41 to Page 8 Line 2 for the revisions.

**Comment NO.3:** *Line 4 on Page 7, point out the criteria of the selected three days, for example similar meteorological conditions and/or pollution levels.*

**Response:** The criteria of the selected three days for Figure 5 is the similar meteorological conditions. The meteorological conditions are unfavorable for the dispersion of atmospheric pollutants on these three days. The weather circulation field is homogenous or weak pressure field, the wind speed near the ground is small, and the atmospheric static stability is high. The average hourly wind speed on 31 October, 9 November and 15 November, 2014 is 1.19, 1.28 and 1.07 m s$^{-1}$ respectively.

**Changes in Manuscript:** We gave the criteria for the selection of the three days in the revised manuscript, please refer to Page 7 Line 20-24 for the changes.

**Comment NO.4:** *Line28-31 on Page 9, in addition to the missing nighttime data and the gradually implemented control policies in the pre-APEC period, do the less favorable meteorological conditions (see Line 5 on Page 6) and the increased energy consumption due to household heating also contribute to the higher decrease in C$_{veh}$ relative to the post-APEC period?*

**Response:** The referee's concerns about other possible reasons contribute to the decrease in C$_{veh.}$ in this comment are responded below:

**(1) Meteorological Conditions.**

Firstly, we have considered the impact of meteorological conditions in the experimental design. All observation trips started at the same period of 1 day (i.e. about 10:00 am LT in the daytime and about 1:00 am LT in the nighttime), with relatively stable

atmospheric boundary layer during the selected observation period, which can eliminate the meteorological influence to some extent. Secondly, direct measurements of wind speed and wind direction from the mobile lab during driving may have large deviations (Johansson, 2009; Wang et al., 2011). Meanwhile the observation data of wind speed and direction at the ground based site (such as the Nanjiao site) is hourly mean data, with Low time resolution and limited spatial representation. It is difficult to obtain high time-resolution and accurate meteorological data matching the concentration of gaseous pollutants. The lack of high time resolution meteorological data makes it difficult to accurately evaluate and eliminate the meteorological impacts. Therefore, we thought that the short-term micro-scale meteorology has little effect on our assessment results, and the discussion and accurate quantification of micro-scale meteorological influences are beyond the scope of this study. In this paper, we assumed that the high-frequency component is confined to much shorter time scales than those of the meteorological factors, and that changes in meteorological conditions were reflected only in the low-frequency component ($C_{bg.}$), such that changes in meteorological conditions did not influence our evaluation of control polices.

**(2) The increased energy consumption due to household heating.**

As we mentioned in the manuscript, the sudden drop in temperature may have led to an increase in heating demand, which could have had an impact on household pollutant emissions (Liu et al., 2016). However, since our observations are at specific time of 1 day, the emission variations of the household heating sources are considered to be relatively stable compared to the vehicle emission sources throughout the post-APEC phase. Thus we thought the changes in energy consumption due to household heating were reflected only in the low-frequency component ($C_{bg.}$). And the increased energy consumption due to household heating do not contribute to the higher decrease in $C_{veh.}$ relative to the post-APEC period.

**Changes in Manuscript:** We added discussion on the impact of meteorological conditions and household heating in the revised manuscript, please refer to Page 6-7 and Page 6 Line 16-29 and Page 10 Line 17-18 for the changes.

[revised manuscript text omitted]
 blue line shows the linear fit curve and the gray area is the 95% confidence interval.

[Figure]

Figure S5. Correlations between the on-road background concentrations (i.e. $C_{bg.}$) obtained by the wavelet transform method (WTM) and observations at the PKU site in each day for NO, $NO_x$, BC, CO, $SO_2$, and $O_3$. The blue line shows the linear fit curve and the gray area is the 95% confidence interval.

**Table S1.** Timing of on-road measurements made around the 4th Ring Road of Beijing.

| Date (MM/DD_Night, Day) | Start time[a] (hh:mm) | End time[b] (hh:mm) | Driving duration[c] (h) |
|---|---|---|---|
| 10/28_Day | 10:17 | 11:55 | 1.63 |
| 10/29_Day | 09:52 | 11:20 | 1.47 |
| 10/30_Day | 20:06 | 21:13 | 1.12 |
| 10/31_Day | 10:09 | 11:28 | 1.32 |
| 11/01_Day | 09:53 | 11:18 | 1.42 |
| 11/02_Day | 09:56 | 11:24 | 1.46 |
| 11/03_Day | 10:05 | 11:17 | 1.20 |
| 11/04_Night | 00:58 | 02:06 | 1.13 |
| 11/04_Day | 09:56 | 11:29 | 1.55 |
| 11/05_Night | 00:58 | 02:12 | 1.23 |
| 11/05_Day | 09:56 | 11:10 | 1.23 |
| 11/06_Night | 01:10 | 02:22 | 1.20 |
| 11/06_Day | 09:58 | 11:10 | 1.20 |
| 11/07_Night | 01:04 | 02:05 | 1.02 |
| 11/07_Day | 09:58 | 11:08 | 1.17 |
| 11/08_Night | 00:56 | 02:05 | 1.15 |
| 11/08_Day | 09:58 | 11:12 | 1.23 |
| 11/09_Night | 00:58 | 02:07 | 1.15 |
| 11/09_Day | 09:58 | 11:07 | 1.15 |
| 11/10_Night | 00:59 | 02:07 | 1.13 |
| 11/10_Day | 09:59 | 11:08 | 1.15 |
| 11/11_Night | 00:59 | 2:09 | 1.17 |
| 11/11_Day | 10:01 | 11:12 | 1.18 |
| 11/12_Night | 00:59 | 02:11 | 1.20 |
| 11/12_Day | 09:58 | 11:06 | 1.13 |
| 11/13_Night | 00:58 | 02:08 | 1.17 |
| 11/13_Day | 09:57 | 11:18 | 1.35 |
| 11/14_Night | 00:58 | 02:08 | 1.17 |
| 11/14_Day | 09:57 | 11:45 | 1.80 |
| 11/15_Night | 00:58 | 02:08 | 1.17 |
| 11/15_Day | 10:00 | 11:28 | 1.47 |
| 11/16_Night | 00:58 | 02:10 | 1.20 |
| 11/16_Day | 10:02 | 11:19 | 1.28 |
| 11/17_Night | 00:58 | 02:08 | 1.17 |
| 11/17_Day | 09:57 | 11:29 | 1.53 |
| 11/18_Night | 00:59 | 02:08 | 1.15 |
| 11/18_Day | 10:03 | 12:03 | 2.00 |
| 11/19_Night | 00:58 | 02:06 | 1.13 |

| | | | |
|---|---|---|---|
| 11/19_Day | 09:57 | 011:16 | 1.32 |
| 11/20_Night | 00:59 | 02:09 | 1.17 |
| 11/20_Day | 09:57 | 11:38 | 1.68 |
| 11/21_Night | 00:58 | 02:10 | 1.2 |
| 11/21_Day | 10:02 | 11:59 | 1.95 |
| 11/22_Night | 00:58 | 02:06 | 1.13 |

[a]Start-time is when the monitoring vehicle entered the 4th Ring Road.

[b]Start-time is when the monitoring vehicle left the 4th Ring Road.

[c]Driving duration is the length of time when the monitoring vehicle driving along the 4th Ring Road.